# Epsin-mediated degradation of IP3R1 fuels atherosclerosis

Yunzhou Dong[1], Yang Lee[1], Kui Cui[1], Ming He[2], Beibei Wang[1], Sudarshan Bhattacharjee [1], Bo Zhu[1], Tadayuki Yago[3], Kun Zhang[1], Lin Deng[4], Kunfu Ouyang[2], Aiyun Wen[1], Douglas B. Cowan [1,5], Kai Song[1], Lili Yu[1], Megan L. Brophy[1], Xiaolei Liu[6], Jill Wylie-Sears[1], Hao Wu[1], Scott Wong[1], Guanglin Cui[7], Yusuke Kawashima [8,9], Hiroyuki Matsumoto[8], Yoshio Kodera[9], Richard J. H. Wojcikiewicz[10], Sanjay Srivastava[11], Joyce Bischoff [1], Da-Zhi Wang[5], Klaus Ley[12] & Hong Chen[1✉]

The epsin family of endocytic adapter proteins are widely expressed, and interact with both proteins and lipids to regulate a variety of cell functions. However, the role of epsins in atherosclerosis is poorly understood. Here, we show that deletion of endothelial epsin proteins reduces inflammation and attenuates atherosclerosis using both cell culture and mouse models of this disease. In atherogenic cholesterol-treated murine aortic endothelial cells, epsins interact with the ubiquitinated endoplasmic reticulum protein inositol 1,4,5-trisphosphate receptor type 1 (IP3R1), which triggers proteasomal degradation of this calcium release channel. Epsins potentiate its degradation via this interaction. Genetic reduction of endothelial IP3R1 accelerates atherosclerosis, whereas deletion of endothelial epsins stabilizes IP3R1 and mitigates inflammation. Reduction of IP3R1 in epsin-deficient mice restores atherosclerotic progression. Taken together, epsin-mediated degradation of IP3R1 represents a previously undiscovered biological role for epsin proteins and may provide new therapeutic targets for the treatment of atherosclerosis and other diseases.

[1] Vascular Biology Program, Boston Children's Hospital, Harvard Medical School, Boston, MA 02115, USA. [2] Department of Medicine, University of California, San Diego, San Diego, CA 92093, USA. [3] Cardiovascular Biology Program, Oklahoma Medical Research Foundation, Oklahoma City, OK 73104, USA. [4] Department of Biological Chemistry and Molecular Pharmacology, Harvard Medical School, Boston, MA 02115, USA. [5] Department of Cardiology, Boston Children's Hospital, Harvard Medical School, Boston, MA 02115, USA. [6] Center for Vascular and Developmental Biology, Feinberg Cardiovascular Research Institute, Feinberg School of Medicine, Chicago, IL 60611, USA. [7] Department of Nutrition and Epidemiology, Harvard T.H. Chan School of Public Health, Boston, MA 02115, USA. [8] Department of Biochemistry and Molecular Biology, University of Oklahoma Health Sciences Center, Oklahoma City, OK 73104, USA. [9] Center for Disease Proteomics, Kitasato University School of Science, 1-15-1 Kitasato, Minami-ku, Sagamihara, Kanagawa 252-0373, Japan. [10] Department of Pharmacology, SUNY Upstate Medical University, Syracuse, NY 13210, USA. [11] Department of Medicine, Division of Cardiovascular Medicine, University of Louisville School of Medicine, Louisville, KY 40202, USA. [12] La Jolla Institute for Allergy and Immunology, La Jolla, CA 92037, USA. ✉email: hong.chen@childrens.harvard.edu

Atherosclerosis is a leading cause of morbidity and mortality in the United States and other developed countries[1]. The progressive accumulation of atherosclerotic plaques, or atheromas, occludes blood flow, causes hardening of the arteries, and results in heart attacks or strokes[1,2]. Atheromas develop in the inner layer of the arterial wall and are predominately composed of mineral deposits, fibrous connective tissues, and lipid-laden macrophages called foam cells[3]. Understanding the molecular mechanisms responsible for the initiation, growth, and destabilization of atheromas is essential for the development of more effective and targeted therapies to prevent ischemic injury, disability, or death in patients with cardiovascular disease.

Chronic inflammation activates various signaling cascades, which, in turn, recruit pro-inflammatory macrophages that infiltrate the inner arterial wall and transition to lipid-laden and cholesterol-laden foam cells[4,5]. Foam cell accumulation and prolonged inflammatory signaling eventually results in formation of a necrotic core that destabilizes atheromas and greatly increases the risk of a thrombotic event[5,6]. Endothelial dysfunction in the artery as a result of injury or abnormal metabolic regulation is an ancillary pro-inflammatory event that facilitates chronic inflammation and macrophage recruitment; however, the underlying molecular interactions that drive endothelial dysfunction and promote pro-inflammatory signaling at the arterial wall remain unclear[7–10]. Given that endothelial dysfunction is one of the early events in the pathogenesis of atherosclerosis, identifying the underlying cellular mechanisms may provide molecular targets for the development of innovative therapies to treat this disease.

Epsins are family of highly-conserved, membrane-associated, ubiquitin-binding endocytic adapter proteins[11–16]. Epsins 1 and 2 are ubiquitously expressed and largely redundant in function, while epsin 3 expression is primarily limited to keratinocytes and the stomach[17]. Deficiency of all three epsins impairs receptor-mediated endocytosis by hindering the actin-dependent invagination of clathrin-coated pits[16]. On the other hand, loss of epsins 1 and 2 does not impair housekeeping membrane trafficking including transferrin receptor endocytosis and endothelial growth factor receptor (EGFR) internalization[18,19]. Using cell-type and tissue-specific epsin knock-out mouse models, we revealed the critical regulatory roles that these proteins play in the progression of a variety of diseases including cancer and atherosclerosis[20–22]. Given the role of chronic inflammation in both cancer and atheroma progression, we hypothesized that inhibition of epsin function may provide a potential therapeutic target for combating atherosclerosis. In support of this notion, we recently reported that deletion of macrophage (MΦ) epsins protected against atherosclerosis by reducing plaque deposition by approximate 50% in a mouse model[22]. Mechanistically, macrophage epsins were found to govern the endocytosis and degradation of low-density lipoprotein receptor-related protein 1 (LRP-1), which is a plasma membrane protein implicated in mediating efferocytosis in macrophages. Epsin-mediated LRP-1 degradation facilitated atheroma progression; however, whether epsins promote endothelial dysfunction through other protein interactions to contribute to the progression of atherosclerosis is unknown.

For that reason, we explored the role of endothelial epsins in exacerbating endothelial dysfunction associated with atherosclerosis. Using a novel endothelial-specific epsin-deficient ApoE$^{-/-}$ atherosclerotic mouse model, we discovered that endothelial-specific depletion of epsin proteins prevented endothelial cell dysfunction and halted atherosclerosis. We established that the ubiquitin-interacting motif (UIM) of epsin 1 physically associates with the ubiquitinated suppressor domain (SD) in the N-terminal region of the endoplasmic reticulum (ER) calcium release channel, inositol 1,4,5-trisphosphate (IP3) receptor 1 (IP3R1). Under atherogenic conditions, we found epsin 1 accelerates ubiquitin-dependent degradation of IP3R1, which leads to aberrant calcium homeostasis, inflammation, and progression of atherosclerosis[23,24]. Consequently, epsins and proteins that interact with these adapters may represent potential therapeutic targets for the treatment of atherosclerosis.

## Results

**Epsin interacts with IP3R1 in endothelial cells treated with atherogenic stimuli.** Because epsins 1 and 2 are paralogs with overlapping expression and function, we focused primarily on epsin 1 for in vitro studies; however, to elucidate the role of endothelial epsins in atherosclerosis, we generated ApoE$^{-/-}$ mice with an endothelial cell-specific deletion of epsin 1 on a global epsin 2 knock-out background (EC-iDKO/ApoE$^{-/-}$) to circumvent the redundant function of two epsins (Supplementary Fig. 1a, b)[18,19,25]. Conditional deletion of epsin 1 on the epsin 2$^{-/-}$ background in aortic ECs was confirmed by immunofluorescent staining of cryopreserved aortic roots (Supplementary Fig. 1c). Aortic tissues and endothelial cells from ApoE$^{-/-}$ (control, see "Methods" section for a detailed explanation) and EC-iDKO/ApoE$^{-/-}$ (iDKO) mice were then used to investigate the expression and function of endothelial epsins, as well as interacting proteins in atherosclerosis initiation and progression.

To detect interactions of the epsin 1 adapter protein with other proteins under atherogenic conditions, we immunoprecipitated binding partners from primary murine aortic endothelial cells (MAECs) treated for 30 min with oxidized low-density lipoprotein (oxLDL) using an epsin 1 antibody and performed mass spectrometry (Fig. 1a, b and Supplementary Data file 1). MAECs were characterized and compared to murine brain endothelial cells (MBECs) to establish that aortic endothelial cells were capable of forming tubular networks (Supplementary Fig. 2a, b) and, unlike brain endothelial cells, expressed low levels of vascular endothelial growth factor 2 (VEGF2) in addition to expressing high levels of von Willebrand factor (vWF), CD31, and VE-cadherin in the absence of alpha-smooth muscle actin (α-SMA) by immunostaining and FACS analysis (Supplementary Fig. 2c-h; Supplementary Fig. 3). MAECs derived from iDKO mice contained negligible quantities of epsin proteins compared to cells isolated from wild-type (ApoE$^{-/-}$) mice, which confirmed the efficiency of the double (epsin 1 and 2) knock-out strategy (Supplementary Fig. 2h). Atherogenic cholesterol-treated MAECs were subsequently used to identify proteins that bind epsin 1.

Intriguingly, epsin 1 immunoprecipitated with inositol 1,4,5-trisphosphate (IP3) receptor 1 (IP3R1) in addition to the predicted endocytic proteins (e.g., clathrin, epsin 2) (Fig. 1b). IP3R1 is the principal calcium release channel located in the membrane of the endoplasmic reticulum (ER) and is important in maintaining calcium homeostasis. As aberrant calcium signaling and ER stress can substantially contribute to endothelial dysfunction and atherogenesis, we investigated the connection between epsin 1 and IP3R1 in cholesterol-treated endothelial cells[26,27]. MAECs treated with oxidized low-density lipoprotein (oxLDL), crystal cholesterol (Chol), or 7-ketocholesterol (7-KC) showed increases in the amount of IP3R1 interacting with epsin 1, regardless of whether epsin 1 or IP3R1 antibodies were used for immunoprecipitation from cell lysates (Fig. 1c–f). As inositol trisphosphate receptors are rapidly ubiquitinated and degraded in response to cell stress, the proteasome inhibitor MG132 was applied to MAECs to impede degradation of ubiquitin-conjugated proteins. Long-term treatment (36 h) of cells with atherogenic cholesterol destabilized IP3R1, which typically decreases in

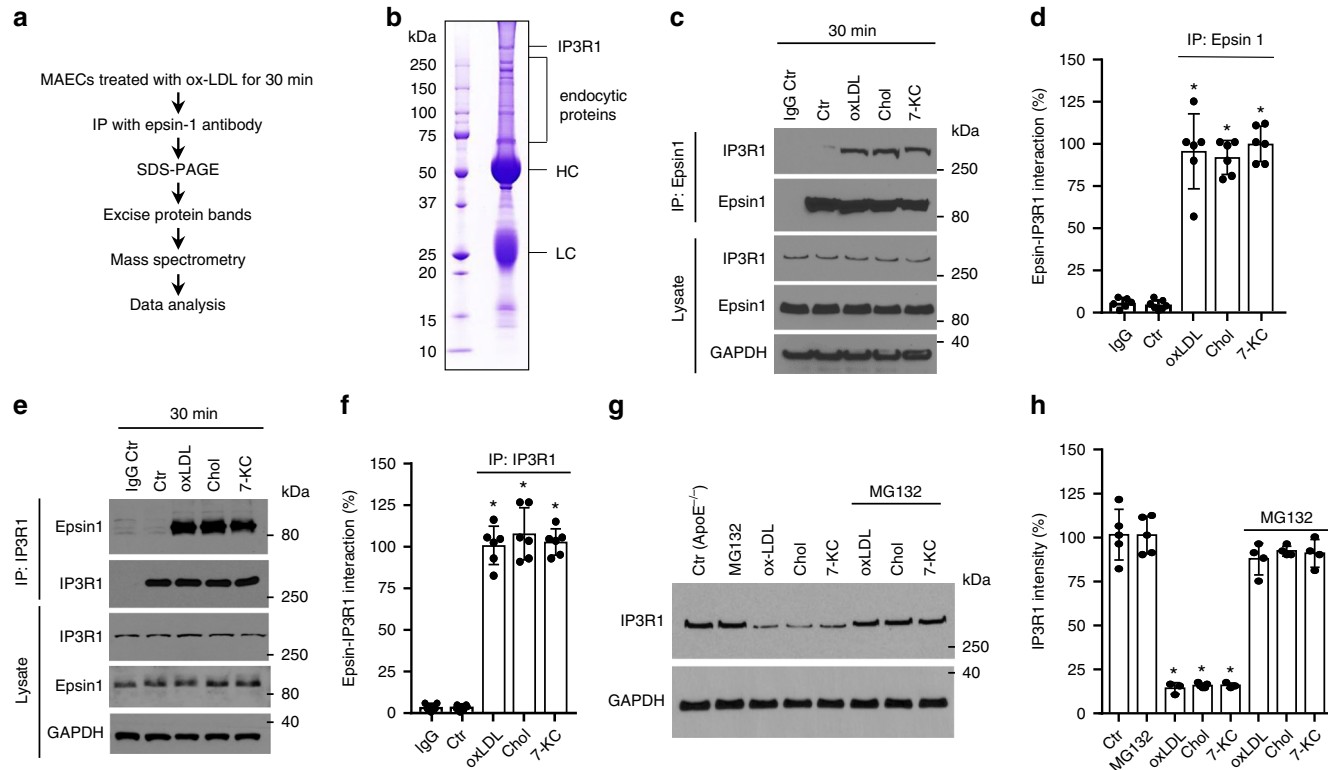

**Fig. 1 IP3R1 interacts with epsin 1 in endothelial cells treated with atherogenic cholesterol. a** Strategy for identifying epsin 1 binding proteins. **b** Coomassie-stained gel of the epsin 1 IP immunocomplex from MAECs. Bands were characterized by mass spectrometry. HC heavy chain, LC light chain. **c**, **d** MAECs isolated from WT mice were treated with 100 μg/mL oxLDL or 500 μg/mL cholesterol crystal (Chol) or 50 μM 7-KC for 30 min and an immunoprecipitation (IP) was performed using the epsin 1 antibody. The immunocomplex supernatant was probed with antibodies to IP3R1 or epsin 1 by western blot analysis (**c**) and then quantified (**d**) (n = 6 independent experiments, asterisk P < 0.05 compared with control). **e**, **f** Reciprocal IP using the IP3R1 antibody in the same samples as described in **c** and **d**. A representative western blot (**e**) and quantification (**f**) (n = 6 independent experiments, asterisk P < 0.05 compared to control) of the IP are shown. **g**, **h** MAECs isolated from control mouse aortas (i.e., from ApoE$^{-/-}$ mice) were pre-treated with 5 μM MG132 for 4 h followed by treatment with 100 μg/mL oxLDL or 500 μg/mL Chol or 50 μM 7-KC for 36 h in the presence of vehicle (DMSO) or MG132. IP3R1 was measured by western blot (**g**) and quantified (**h**) (n = 5 independent repeats, asterisk P < 0.001 compared to control). All data were assessed using Student's t-test and are presented as the mean ± SEM.

chronic atherosclerosis; however, MG132 treatment reverses this decrease suggesting this receptor is targeted for degradation in atherosclerosis (Fig. 1g, h)[28,29]. Although IP3R1 degradation through ubiquitin-dependent ER-associated degradation (ERAD) is well documented[30,31], we also wanted to determine whether other proteolytic systems were involved in oxLDL-mediated IP3R1 degradation. Our results show that biochemical or genetic inhibition of lysosomal proteolysis, the unfolded protein response (UPR), and autophagy had little to no effect on IP3R1 degradation (Supplementary Fig. 4a, b). At the same time, we cannot rule out other forms of protein degradation, since chemical inhibitors are not exclusively specific; so, further research of IP3R1 degradation pathway(s) is warranted using genetically engineered models.

Immunofluorescent staining of IP3R1 in aortas from human atherosclerotic patients verified the reduction in receptor expression through the course of disease (Supplementary Fig. 5a, b). Moreover, immunostaining of control (ApoE$^{-/-}$) mice fed a WD showed less IP3R1 in the endothelium and medial layers of the aorta compared to control animals fed a regular chow diet or iDKO (EC-iDKO/ApoE$^{-/-}$) mice fed a WD (Supplementary Fig. 5c, d). These results substantiate the inverse correlation between IP3R1 levels and atherosclerosis.

**IP3R1 SD and epsin 1 UIM domains interact under atherogenic conditions**. To better understand the molecular interaction

between IP3R1 and epsin 1, we mapped protein binding domains by transfecting HEK 293T cells with a variety of pcDNA3 expression constructs. Hemagglutinin (HA)-tagged IP3R1 constructs and FLAG epitope-tagged epsin 1 constructs containing precise deletions of the N-terminal homology (ENTH) domain and/or the ubiquitin-interacting motif (UIM) were expressed in HEK 293T cells for 24 h and used for immunoprecipitation and western blot analyses with antibodies to each epitope tag, as well as specific proteins (Fig. 2a–c and Supplementary Fig. 6a, b). Our results showed deletion of the ENTH domain had only a minor effect on the epsin 1-IP3R1 interaction relative to deletion of the UIM, while the clathrin-binding DPW motif and the asparagine-proline-phenylalanine (NPF) motif did not substantially contribute to the epsin 1-IP3R1 interaction. These findings implicate the epsin 1 UIM as the obligatory site for interaction with IP3R1.

Alternatively, mapping of the epsin 1 binding site to expressed, truncated IP3R1 constructs in HEK 293T cells revealed the N-terminal domain (NTD) of IP3R1 was critical for the epsin 1-IP3R1 interaction (Fig. 2d–f). Additional fine mapping experiments with expression constructs containing precise deletions in the NTD of IP3R1 demonstrated the suppressor domain (SD), which decreases affinity to the inositol triphosphate (IP3)-binding cores (IBCα and IBCβ), was essential for epsin 1 binding (Fig. 2g–i and Supplementary Fig. 6c). As a result, our detailed mapping experiments show the UIM in epsin and the SD in IP3R1 were responsible for the molecular interaction of these proteins.

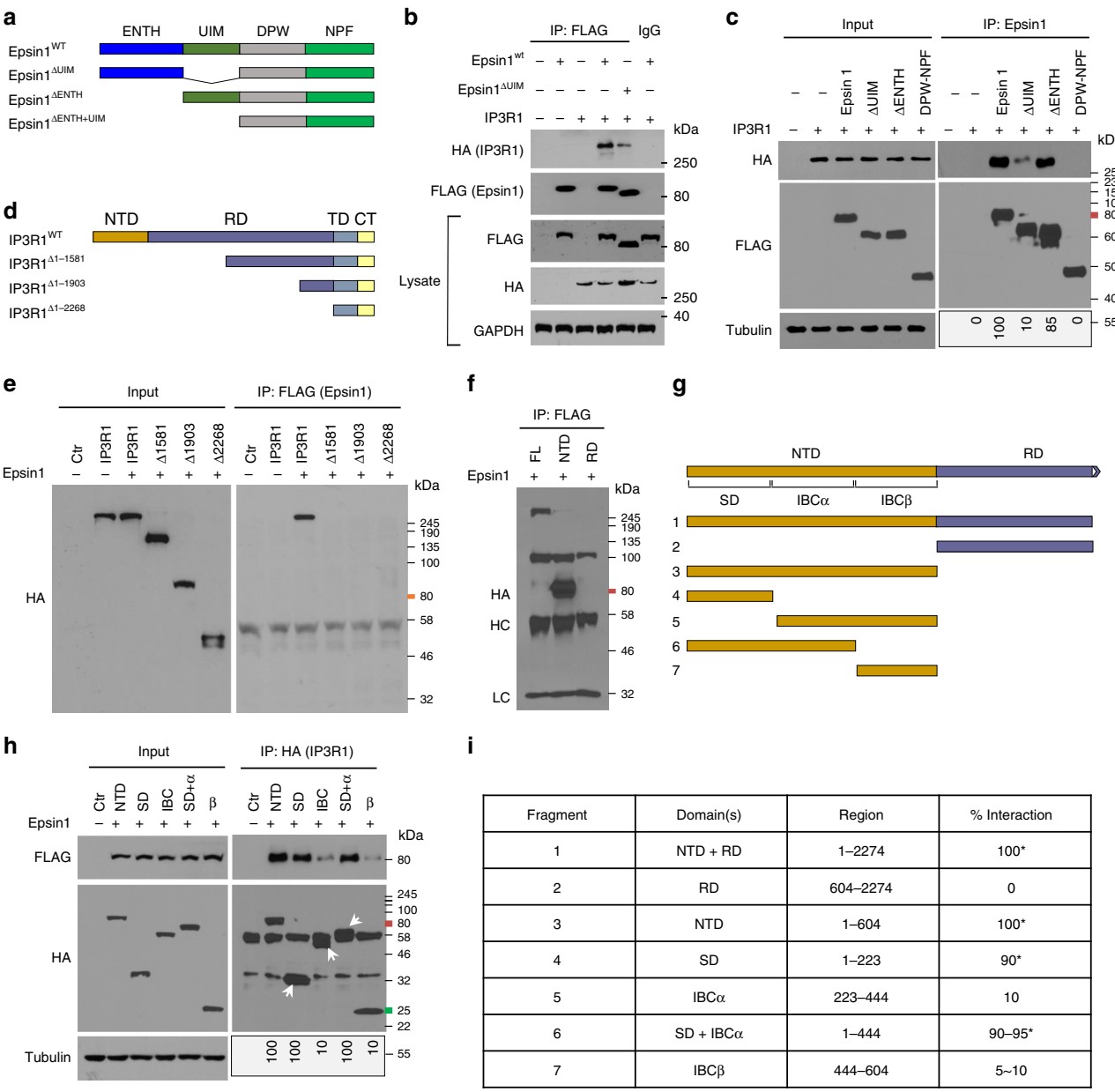

**Fig. 2 IP3R1 SD and epsin 1 UIM domains interact under atherogenic conditions. a** to **c** Full length and domain-deletion constructs of FLAG-epsin 1 in the pcDNA3 vector were transfected into HEK 293T (293tsA1609neo) cells for 24 h, followed by IP and western blot analysis using antibodies against HA (IP3R1) or FLAG (epsin 1) tags, as well as specific proteins ($n = 5$). **d–f** Full length and deletion constructs of HA-IP3R1 were transfected into HEK 293T cells for 24 h, followed by IP and western blot analysis using antibodies against HA or FLAG tags and specific proteins ($n = 5$). **g** to **i** Fine mapping of the N-terminus of IP3R1 to determine the region that interacts with epsin 1. Interaction results are presented as a 0–100 scale mapping domain(s) that bind(s) epsin 1 upon co-expression in 293T cells ($n = 6$, asterisk $P < 0.001$, compared to the fragments deleted SD). Transfected HEK 293T cells were all treated with 100 μg/mL oxLDL for 30 min. NTD N-terminal domain, RD regulatory domain, TMD transmembrane domain, CTT C-terminal cytoplasmic tail, SD suppressor domain, IBC IP3 binding core composed of α and β domains.

**Epsin 1 modulates receptor degradation through binding to ubiquitinated IP3R1**. To establish the consequences of the interaction between epsin 1 and IP3R1, we isolated MAECs from control (ApoE$^{-/-}$) and mutant mice (EC-iDKO/ApoE$^{-/-}$) and treated these cells with atherogenic cholesterol to determine the effect on specific gene expression, as well as ubiquitination and proteasomal degradation of IP3R1. Unlike acute (30 min) treatments with 100 μg/mL oxLDL or 50 μM 7-KC (Fig. 1c–f), total cell lysates from control mouse endothelial cells subjected to 36 h of stimulation showed significant increases in both epsin 1 and 2 protein levels (Fig. 3a, b). MAECs from EC-iDKO/ApoE$^{-/-}$ (iDKO) mice treated with cholesterol for 36 h did not exhibit

changes in IP3R1 mRNA levels, implying IP3R1 was regulated by epsins at the protein level, which is consistent with the corresponding GO analyses (Supplementary Fig. 6d and, as well as Supplementary Data file 2). In support of this interpretation, acute oxLDL treatment (0, 50, 100, and 200 μg/mL) increased ubiquitination of IP3R1 in the presence of MG132, which inhibits degradation of ubiquitin-conjugated proteins and permits their quantitation (Fig. 3c and Supplementary Fig. 6f). The increased ubiquitination of IP3R1 in response to long-term atherogenic cholesterol results in proteasomal degradation and loss of this calcium release channel from the ER membrane (Fig. 1g, h).

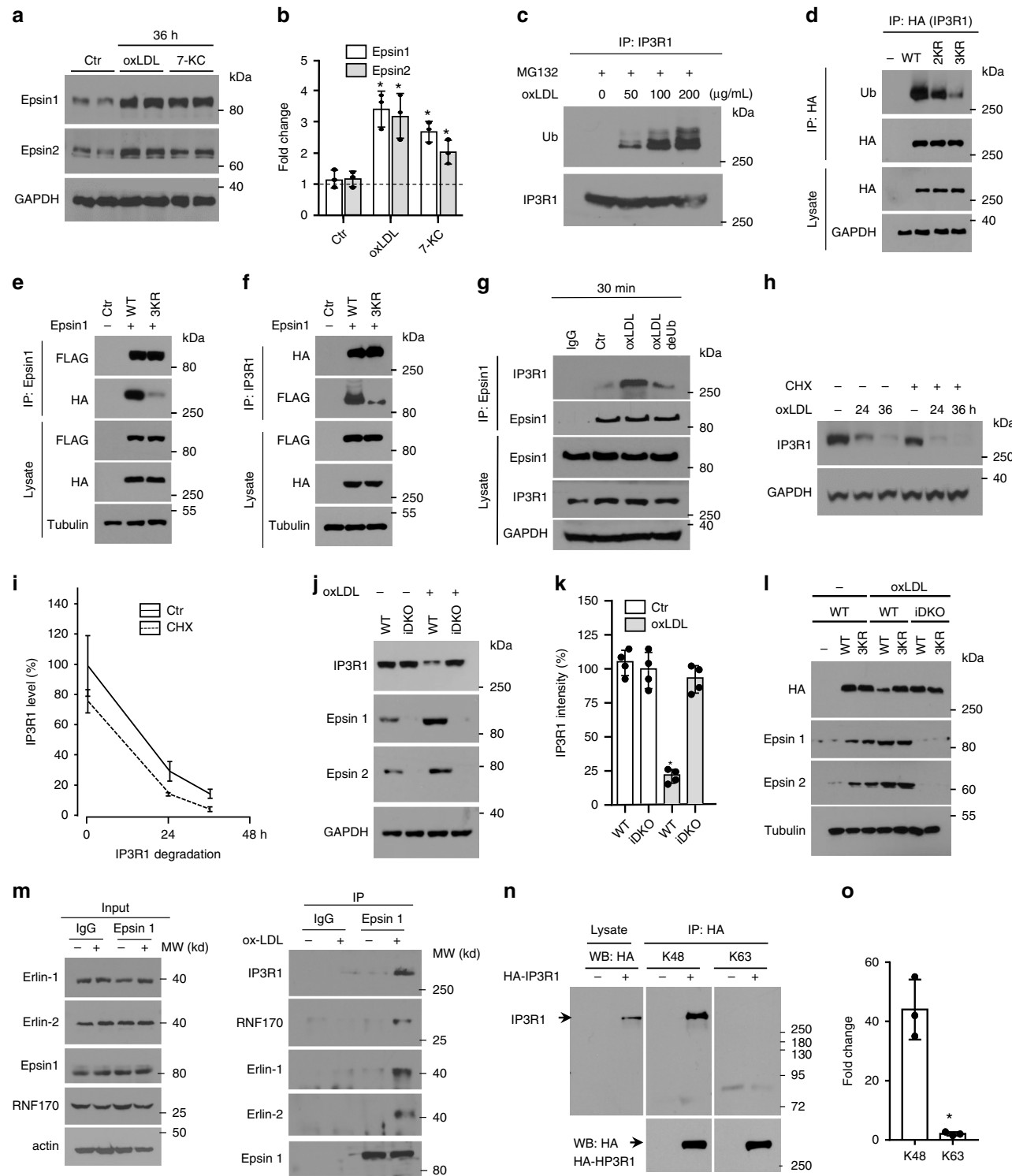

For that reason, we performed bioinformatic analyses to predict the sites of ubiquitination in the IP3R1 SD as this region was essential for epsin 1 binding (Supplementary Fig. 7a). Expression of IP3R1 with lysine (K) to arginine (R) substitutions in putative sites using HEK 293T cells followed by immunoprecipitation of lysates with an antibody to the HA-tag demonstrated double K126R/K129R (2KR) and triple K126R/K129R/K143R (3KR) mutations in the SD impeded ubiquitination of IP3R1 (Fig. 3d and Supplementary Fig. 7b). In addition, we ascertained the molecular mechanism by which epsin 1 regulates IP3R1 degradation through the ubiquitin-proteasome pathway. First, we

transfected FLAG-tagged epsin 1 with the HA-tagged 3KR IP3R1 construct in HEK 293T cells treated with oxLDL for 30 min and then performed reciprocal immunoprecipitations on cell lysates. Western blot analyses of whole cell lysates and immunoprecipitates confirmed IP3R1 with K to R mutations in amino acid positions 126, 129, and 143 attenuated the epsin 1 interaction (Fig. 3e, f; Supplementary Fig. 7c, d). Second, we treated wild-type MAECs with 100 μg/mL oxLDL for 30 min and immunoprecipitated epsin 1 interacting proteins from lysates in the presence or absence of 10 μM USP2, which is a recombinant deubiquitinase (deUb). Western blots showed decreasing ubiquitination of IP3R1

**Fig. 3 Epsin 1 modulates IP3R1 degradation through binding to ubiquitinated IP3R1. a**, **b** Epsin expression in MAECs treated with 100 µg/mL oxLDL or 50 µM 7-KC (36 h). Western blot analysis of epsins (**a**) and quantification (**b**) ($n = 3$, asterisk $P < 0.001$). **c** MAECs treated with oxLDL (30 min) in the presence of 5 µM MG132 followed by IP and western blotting for ubiquitin and IP3R1 ($n = 5$). **d** HA-tagged IP3R1 containing K126R/K129R (2KR) and K126R/K129R/K143R (3KR) mutations transfected into 293T cells (24 h) were treated with 100 µg/mL oxLDL (30 min), then used for IP and western blot analysis for HA (IP3R1), ubiquitin, and proteins ($n = 4$). **e**, **f** HA-tagged IP3R1 containing 3KR mutations and FLAG-epsin 1 transfected into 293T cells (24 h), were treated with 100 µg/mL oxLDL (30 min), and used for IP and western blot analysis for HA, FLAG, and proteins ($n = 4$). **g** MAECs treated with 100 µg/mL oxLDL (30 min) followed by IP and western blot analysis for epsin 1. Lysates were treated with 10 µM of deubiquitinase (deUb) USP2 ($n = 4$, asterisk $P < 0.05$ vs. oxLDL). **h**, **i** MAECs treated with 100 µg/mL oxLDL for 0, 24, and 36 h ± 10 µg/mL cyclohexamide (CHX). Western analysis for IP3R1 (**h**) and quantification (**i**) ($n = 5$). **j**, **k** MAECs isolated from ApoE$^{-/-}$ (WT) or EC-iDKO/ApoE$^{-/-}$ (iDKO) mice treated with 100 µg/mL oxLDL (36 h) were analyzed by western blotting for IP3R1 (**j**) and quantified (**k**) ($n = 4$, $P < 0.001$). **l** HA-tagged wild-type (WT) or 3KR IP3R1 transfected control (WT) or iDKO MAECs were treated with 100 µg/mL oxLDL (30 h). Lysates were analyzed by western blotting for HA (IP3R1), epsins, and tubulin ($n = 5$). **m** MAECs treated ±100 µg/mL oxLDL (4 h) in the presence of 10 µM MG132, followed by epsin 1 IP and immunoblotting ($n = 3$). **n** HA-tagged IP3R1 transfected 293T cells were treated with 100 µg/mL oxLDL (4 h) in the presence of 10 µM MG132. HA immunoprecipitated lysates were analyzed for K48 or K63 ($n = 3$, asterisk $P < 0.0001$). Data was assessed using Student's t-test and presented as mean ± SEM. **o** Quantification of K48 or K63 ubiquitination of transfected IP3R1 ($n = 3$. asterisk $P < 0.001$).

in the presence of USP2 with concomitant reduction in epsin 1 binding, suggesting that ubiquitination is required for epsin 1-IP3R1 interaction (Fig. 3g and Supplementary Fig. 7e). Third, the presence of 10 µg/mL cyclohexamide (CHX) in MAEC cultures treated with 100 µg/mL oxLDL for 0, 24, and 36 h showed IP3R1 degradation paralleled that found in stimulated endothelial cells cultured without translational inhibition, indicating the proteasome catabolized both existing and newly-synesized receptors equally and oxLDL has minimal impact on IP3R1 protein synthesis (Fig. 3h, i). Fourth, epsin 1 and 2 deficiency prevented chronic oxLDL-induced reduction of IP3R1 protein in MAECs isolated from EC-iDKO/ApoE$^{-/-}$ (iDKO) mice, reflecting oxLDL-induced IP3R1 degradation is epsin-dependent (Fig. 3j, k). Finally, western blot analyses of protein lysates derived from wild-type or mutant MAECs transfected with IP3R1 or the 3KR mutant treated with oxLDL for 30 h revealed that the interaction of epsins with ubiquitinated IP3R1 is required for receptor degradation (Fig. 3l and Supplementary Fig. 7f). Combined, these data show ubiquitination of the IP3R1 suppressor domain is obligatory for oxLDL-induced epsin 1 binding and the resultant degradation of this receptor.

Because it has been reported that IP3R1 can be ubiquitinated by the E3 ubiquitin-protein ligase Ring Finger Protein 170 (RNF170)[30], we sought to determine if epsins were associated with this enzyme. We treated MAECs with oxLDL in the presence of MG132 and performed an immunoprecipitation experiment with the epsin 1 antibody. Our results showed that epsin 1 was associated with the RNF170 complex, which is composed of RNF170, erlin 1, and erlin 2. These results indicate that epsins may be involved in the IP3R1 interaction with RNF170 following oxLDL treatment (Fig. 3m). We then overexpressed IP3R1 (with a HA-tag) in HEK 293T cells in the presence of MG132 and oxLDL, and used the HA antibody to immunoprecipitate IP3R1. By subsequently immunoblotting with K48 and K63 antibodies, we discovered that only the K48 antibody produced a band corresponding to IP3R1, while the K63 antibody showed no such band (Fig. 3n, o). These results suggest that IP3R1 is polyubiquitinated with K48-linked chains and subjected to ER-associated proteasomal degradation following atherogenic stimulation.

**Epsin deficiency maintains calcium homeostasis in atherogenic conditions and dampens inflammation in endothelial cells.** To determine the pathophysiological significance of epsin-mediated IP3R1 degradation, we examined aortic endothelial cells treated with cholesterol for signs of impaired regulation of calcium homeostasis in addition to the expression of inflammatory markers. As IP3R1 mediates release of intracellular calcium from the

ER, we stimulated control and iDKO MAECs with increasing concentrations of ATP and recorded intracellular calcium signals using a fluorescence-based method[23,32,33]. While loss of epsins 1 and 2 (iDKO) did not affect intracellular calcium concentrations compared to control (ApoE$^{-/-}$) endothelial cells under basal or resting conditions (Fig. 4a), treatment with oxLDL or 7-KC markedly reduced cytosolic calcium release in control, but not in epsin-deficient, cells (Fig. 4b, c). These data further support the notion that deletion of epsin proteins stabilizes IP3R1, which preserves intracellular calcium homeostasis under atherogenic conditions.

To assess inflammation in stimulated aortic endothelial cells lacking epsins 1 and 2, wild-type or epsin-deficient primary MAECs were treated with 50 ng/mL TNFα for various times to study P-selectin, E-selectin, ICAM-1, and VCAM-1 expression because these proteins mediate leukocyte extravasation[34,35]. As expected, TNFα stimulated all four of these proteins in control cells at either 3 h (P-selectin and E-selectin) or 16 h (VCAM-1 and ICAM-1) (Fig. 4d-g). Conversely, deletion of epsins significantly dampened the induction of these adhesion molecules (Fig. 4d-g). The functional effect of the reduction in selectin and adhesion molecule expression was a significant decline in neutrophil adhesion to TNFα-treated endothelial cells (Fig. 4h, i). We observed a similar reduction in mouse macrophage adhesion. To complement these findings, we performed in vitro macrophage migration assays on the MAECs isolated from WT and EC-iDKO mice. After 3 h of stimulation by LPS or TNFα, our data demonstrate that loss of epsins 1 and 2 significantly reduced macrophage migration through a confluent endothelial cell layer (Fig. 4j, k; Supplementary Fig. 8). As disturbed calcium homeostasis promotes inflammation, our results underscore the importance of epsins in promoting endothelial dysfunction and inflammation in response to atherogenic stimuli in part due to their role in expediting IP3R1 degradation resulting in compromised calcium homeostasis[36].

**Deletion of endothelial epsins 1 and 2 inhibits atherosclerosis.** To understand the in vivo role of epsins in facilitating atherosclerosis through endothelial cell dysfunction, we employed the ApoE$^{-/-}$ atherosclerotic mice with an endothelial cell-specific deletion of epsin 1 on a global epsin 2 knock-out background (EC-iDKO/ApoE$^{-/-}$) (Supplementary Fig. 1a-c)[18,19,25]. To accelerate atherogenesis, ten-week-old mice were fed a western diet (WD) for 8 to 16 weeks. Atherosclerosis was monitored by Oil Red O (ORO) staining of neutral triglycerides deposited in aortic roots and arches from WD-fed ApoE$^{-/-}$ and EC-iDKO/ApoE$^{-/-}$ mice using cryopreserved heart sections or en face preparations[37]. ApoE$^{-/-}$ mice developed progressive lipid deposition

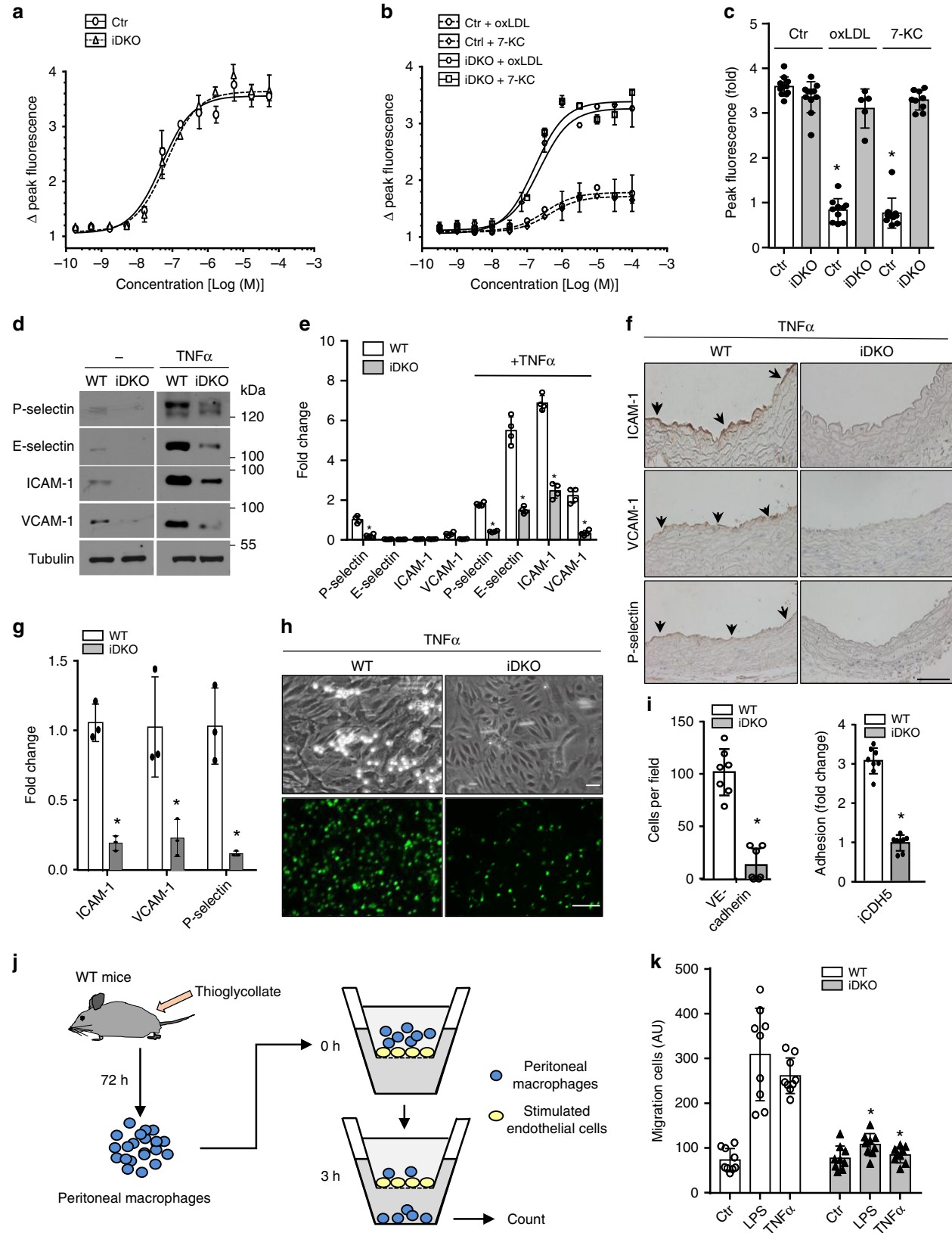

in the aortic root and arch when fed a WD, while loss of endothelial epsins attenuated lipid deposition (Fig. 5a–d and Supplementary Fig. 9a–d). The reduction in atherogenesis resulting from epsin deficiency was comparable in male and female mice through time (not shown). Consistent with the redundant function of epsins 1 and

2, epsin $2^{-/-}$/ApoE$^{-/-}$ mice fed a WD showed no difference in aortic lipid deposition levels when compared with ApoE$^{-/-}$ mice (not shown)[22].

Deletion of endothelial epsins 1 and 2 impaired macrophage (Fig. 5f, g) recruitment to the aorta as measured by immunofluorescent

**Fig. 4 Epsin deficiency maintains calcium homeostasis in atherogenic conditions and dampens inflammation in endothelial cells. a–c** MAECs from control and iDKO mice were stimulated with ATP in the absence (**a**) or presence (**b**) of 100 μg/mL oxLDL or 50 μM 7-KC for 36 h and intracellular calcium was measured and quantified (**c**) ($n = 5$ independent repeats, asterisk $P < 0.001$). **d, e** WT or iDKO MAECs were treated with 50 ng/ml TNFα for 3 h to detect P-selection or E-selectin for 16 h to detect adhesion molecules ICAM-1 and VCAM-1, as well as tubulin by western blot (**d**) prior to quantification (**e**) ($n = 5$ independent repeats, asterisk $P < 0.05$). **f, g** WT and iDKO mice were injected with 0.5 μg TNFα and sacrificed after 3 h (for P-selectin) or 16 h (for adhesion molecules). Heart sections were stained with antibodies for ICAM-1, VCAM-1, and P-selectin. Arrows indicate endothelial staining (**f**), which was quantified (**g**) ($n = 5$ mice in each group, asterisk $P < 0.001$). Scale bar 100 μm. **h, i** Neutrophil adhesion under flow (top panels) and static conditions (bottom panels) to TNF-treated MAECs isolated from WT and iDKO mice (**h**) and quantification (**i**) ($n = 3$ independent repeats, asterisk $P < 0.001$). MAECs were isolated from Epsin 1$^{fl/fl}$/Epsin 2$^{-/-}$: VE-cadherin-Cre (top panels) or Epsin 1$^{fl/fl}$/Epsin 2$^{-/-}$: iCDH5-Cre mice (bottom panels). Scale bars 10 μm (brightfield) or 200 μm (fluorescence). **j, k** Peritoneal macrophage migration through confluent MAECs in the absence or presence of epsin 1 and 2 using a Transwell plate. LPS (200 ng/mL) or TNFα (50 ng/mL) was added to the upper Transwell chamber to stimulate endothelial cells for 3 h, followed by the addition of WT peritoneal macrophages (**j**). After 3 h, macrophages that migrated to the bottom of the Transwell chambers were subjected to enumeration and quantification (LPS and TNFα WT vs. iDKO, $n = 3$ independent experiments, and each treatment selected 3 fields for statistical analysis, asterisk $P < 0.0001$) (**k**). All data were assessed using Student's t-test and are presented as the mean ± SEM.

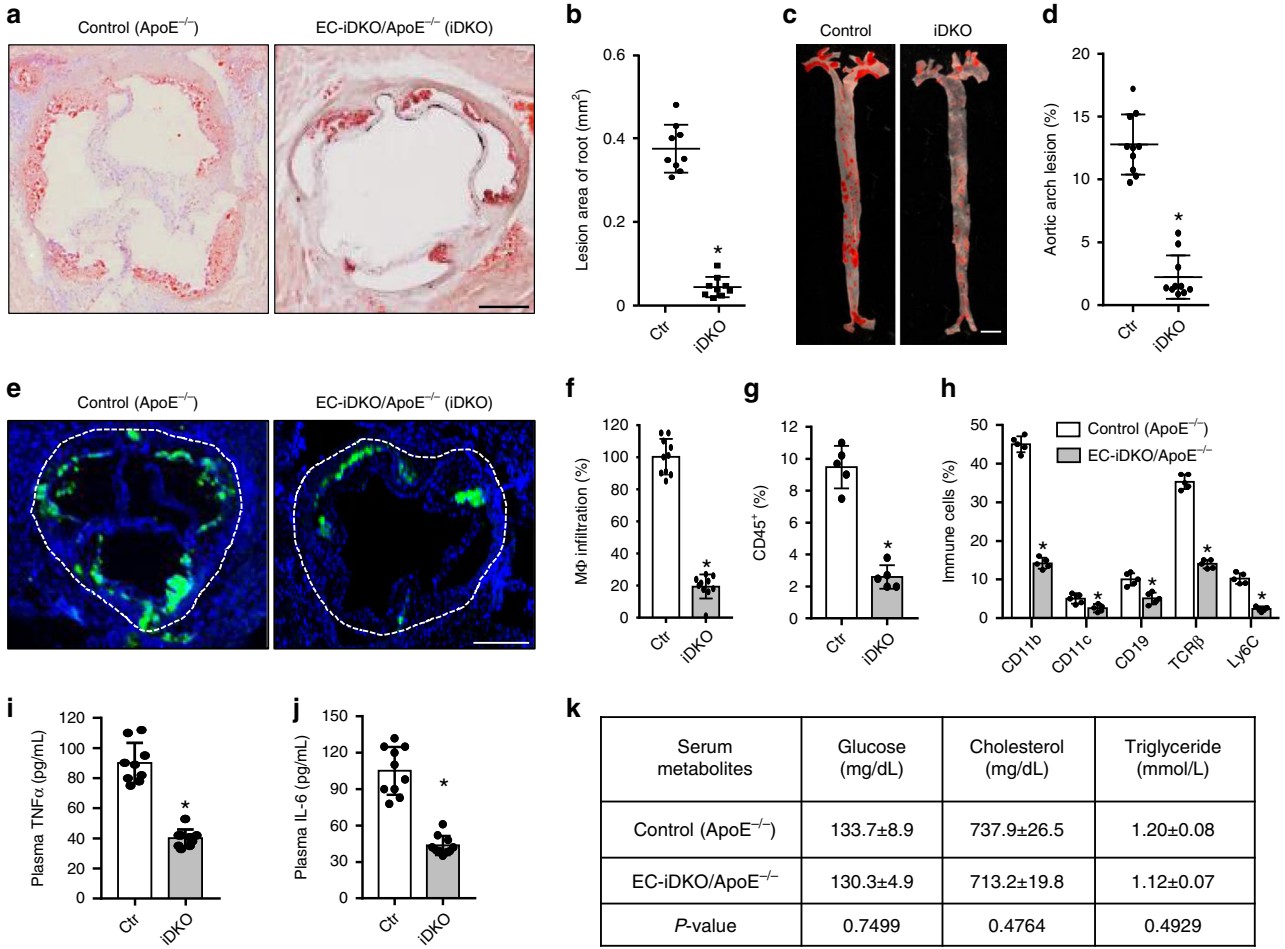

**Fig. 5 Deletion of endothelial epsin 1 and 2 inhibits atherosclerosis. a–d** Oil Red O (ORO) staining of atherosclerotic lesions in aortic roots (**a**) and arches (**c**) of control (ApoE$^{-/-}$) and EC-iDKO/ApoE$^{-/-}$ (iDKO) atherosclerotic mice fed a WD for 12–14 weeks. Statistical analysis of aortic root and arch lesions are shown in (**b**) and (**d**), respectively ($n = 9$ mice for aortic root, and $n = 10$ mice for aortic arch analysis, asterisk $P < 0.001$). Scale bars 500 μm (**a**) and 5 mm (**c**). **e, f** Macrophage (MΦ) infiltration in aortic roots by immunofluorescent staining with the Moma-2 antibody for control and iDKO mice fed a WD (**e**) and quantified (**f**) ($n = 10$ in each group, asterisk $P < 0.001$). Scale bar 500 μm. **g** Immune and inflammatory cells in aortic arches were measured using FACS analysis for CD45$^+$ cells. Data are presented as percentage of the total cells analyzed ($n = 5$ in each group, asterisk $P < 0.05$). **h** Percentage of subtypes of immune cells in aortic arches measured by FACS for control and iDKO mice ($n = 5$ mice in each group, asterisk $P < 0.001$). **i, j** Serum TNFα and IL-6 levels in control and iDKO mice ($n = 10$ in each group, asterisk $P < 0.001$). **k** Metabolic parameters in serum ($n = 10$ in each group). Statistical analyses were performed using Student's t-test and are presented as the mean ± SEM.

staining and flow cytometry, respectively. Immune cell subtypes identified from the leukocyte common antigen (CD45$^+$) population collected from the aortas of ApoE$^{-/-}$ and EC-iDKO/ApoE$^{-/-}$ mice included CD11b, CD11c, CD19, T-cell receptor beta (TCRβ),

and LyC-6 positive cells (Fig. 5h). These cells were decreased as a result of epsin-deficiency. Circulating pro-inflammatory cytokines (TNFα and IL-6) were also reduced in EC-iDKO/ApoE$^{-/-}$ mice (Fig. 5i, j). Importantly, the loss of endothelial epsins did not

change whole-body glucose, lipid profiles, or circulating leukocytes (Fig. 5k). To substantiate these findings, we assessed ApoE$^{-/-}$ mice fed normal chow or a WD and confirmed that epsin 1 was increased in the aortic endothelium (Supplementary Fig. 10a). In parallel, we stained aortic arches obtained from human atherosclerotic patients and again found epsin 1 was increased (Supplementary Fig. 10b). Together, these results show that endothelial epsins are upregulated in the atheroprone endothelium and that the deletion of these adapter proteins inhibits atherosclerosis.

**Distribution and abundance of IP3R1 in mouse endothelium and aortas.** As distinct aortic regions are exposed to differential shear stress, we reasoned that IP3R1 levels may vary along the length of these blood vessels. Whole mount staining of aortas and confocal imaging showed that more IP3R1 protein is present in the endothelium of the thoracic aorta relative to the aortic arch and abdominal aorta (Fig. 6a–c). To verify these findings, seven of these aortic regions were used for RNA extraction (see Fig. 6b). Quantitative RT-PCR analyses demonstrated a significantly higher level of IP3R1 expression in the thoracic aorta compared to the aortic arch and abdominal aorta (Fig. 6d). These data suggested that IP3R1 is increased in atheroresistant regions, which are largely subjected to pulsatile shear stress, when compared to atheroprone regions, which are primarily characterized with disturbed hemodynamic forces such as oscillatory shear stress. These data imply that IP3R1 is atheroprotective. To investigate if other isoforms of IP3 receptors (i.e., IP3R2 and IP3R3) would show similar responses, we analyzed all three isoforms by RT-PCR in aortic segments. Our results show that IP3R1 was the predominantly expressed aortic isoform (Fig. 6e). This finding was essentially confirmed in mouse primary aortic endothelial cells (MAECs) (Fig. 6f), which corroborates an earlier report[38].

**Deficiency of endothelial IP3R1 accelerates atherosclerosis.** To interrogate the in vivo role of endothelial IP3R1 in regulating atherosclerosis, we established a conditional endothelial-specific IP3R1 depletion mouse by crossing IP3R1$^{fl/fl}$ mice to ApoE$^{-/-}$ mice followed by breeding of offspring with iCDH5-Cre transgenic mice (Supplementary Fig. 11a–c)[39]. IP3R1-specific deletion by tamoxifen injection in the endothelium was confirmed by western blot analysis in isolated MAECs and by in vivo immunofluorescent staining (Supplementary Fig. 11d, e). When EC-IP3R1-iKO/ApoE$^{-/-}$ mice were fed a WD for 12–14 weeks the aortic roots developed severe atheroma and lipid streaks along the vessel walls (Fig. 7a, b). Atherosclerotic lesions in aortic arches of WD-fed EC-IP3R1-iKO/ApoE$^{-/-}$ were also significantly larger than WD-fed ApoE$^{-/-}$ mice (Fig. 7c, d). In addition, the atherosclerotic lesions of these mice developed large necrotic cores after WD feeding for 25 weeks, suggesting that the loss of endothelial IP3R1 adversely affects atheroma stability (Supplementary Fig. 12a, b). To corroborate these findings, we explored IP3R1 expression in cultured human aortic endothelial cells (HAECs) exposed to hemodynamic forces known to be either atheroprotective or atheroprone. Compared with atheroprone oscillatory shear stress (OS, $1 \pm 4$ dynes/cm$^2$), atheroprotective pulsatile shear stress (PS, $12 \pm 4$ dynes/cm$^2$) significantly induced and increased IP3R1 mRNA and protein levels (Fig. 7e–g). In the absence of epsins 1 and 2, IP3R1 is relatively abundant in both PS and OS conditions (Fig. 7h, i), and, again, OS decreased IP3R1 faster in the presence of epsins 1 and 2, which further indicates that these adapter proteins accelerate IP3R1 degradation (Fig. 7h, i). These data show that endothelial IP3R1 is upregulated by

pulsatile shear stress, which is consistent with the atheroprotective role of IP3R1 in vivo.

**Reduction of endothelial IP3R1 in EC-iDKO restores atherosclerosis.** To examine whether epsins regulate atherosclerosis progression by controlling IP3R1 degradation in endothelial cells in vivo, we crossed EC-iDKO/ApoE$^{-/-}$ mice with EC-IP3R1$^{fl/+}$/ApoE$^{-/-}$ mice to genetically eliminate one IP3R1 allele and determine if we could reverse the atheroprotective effect of epsin-deficiency on lesion progression due to a lessening of degradation of IP3R1 in epsin EC-iDKO mice. Since aortic arch lesions in mice fed a WD for a longer period of time are more stable, we chose to feed mice a WD for 26–28 weeks. We found that EC-iDKO/EC-IP3R1$^{fl/+}$/ApoE$^{-/-}$ mice displayed a phenotype with aortic atherosclerotic plaques that were larger than that of EC-iDKO/ApoE$^{-/-}$ mice, approximate 60–70% of the size of plaques in ApoE$^{-/-}$ mice, suggesting that endothelial IP3R1 heterozygosity restores atherosclerosis in epsin-deficient mice and that the atheroprotective effect of epsin deletion was mainly resulted from IP3R1 stabilization by prevention of proteasomal degradation (Fig. 7h, i). Therefore, our findings reveal that loss of epsins in the endothelium inhibited inflammation and alleviated atherosclerosis through IP3R1 stabilization (Fig. 7j).

**Discussion**

We show that epsin 1 interacts with the ER calcium release channel IP3R1 during atherogenic stimulation in endothelial cells, indicating a dynamic interaction between proteins associated with the plasma membrane and the endoplasmic reticulum (Fig. 1)[40,41]. IP3R1 has a long N-terminal domain and a short C-terminal domain that both face the cytosol[23,42]. Recent discoveries demonstrate that the ER membrane and the plasma membrane dynamically interact and closely associate with one another[40,41], suggesting that epsins, which are traditionally thought of as adapter proteins associated with the plasma membrane, have the potential to interact with IP3R1. The binding of these proteins is facilitated by the ubiquitin-interacting motif (UIM) of epsin 1 and the ubiquitinated suppressor domain (SD) in the N-terminus of IP3R1 (Fig. 2). This interaction results in proteasomal degradation of IP3R1 (Fig. 3), which causes a homeostatic imbalance in the cytoplasmic free calcium concentration exacerbating endothelial dysfunction through inflammation leading to progression of atherosclerosis (Fig. 7). Using ApoE-null atherosclerotic mice (ApoE$^{-/-}$) fed a high cholesterol-high fat western diet (WD) with inducible endothelial-specific deletion of epsins 1 and 2 (EC-iDKO/ApoE$^{-/-}$), we demonstrated that loss of endothelial epsins prevented atheroma lesion growth in the aorta, suggesting that epsins are important regulators of atherosclerosis (Fig. 5). In contrast, loss of IP3R1 in ApoE$^{-/-}$ mice fed a WD caused an increase in vulnerable atherosclerotic plaque formation (Fig. 7). More importantly, genetic removal of one IP3R1 allele in atherosclerotic mice also lacking epsins (iDKO/IP3R1$^{fl/+}$) restores atheroma formation, reflecting the in vivo interaction of epsins and IP3R1. Collectively, our data reveal a role for epsins in promoting endothelial dysfunction and atherosclerosis by controlling ubiquitinated IP3R1 turnover through ER-associated degradation[43–45]. These findings may provide a foundation for the development of therapeutics to treat atherosclerosis.

Inositol 1,4,5-trisphosphate receptors are a family of calcium release channels located in the endoplasmic reticulum, resulting in calcium release from the ER[23,46]. While not completely resolved, recent evidence indicates that deletion of IP3R1 in adult endothelial cells does not affect vasodilation or blood pressure;[46] however deletion of IP3 receptors 1, 2, and 3 in combination from

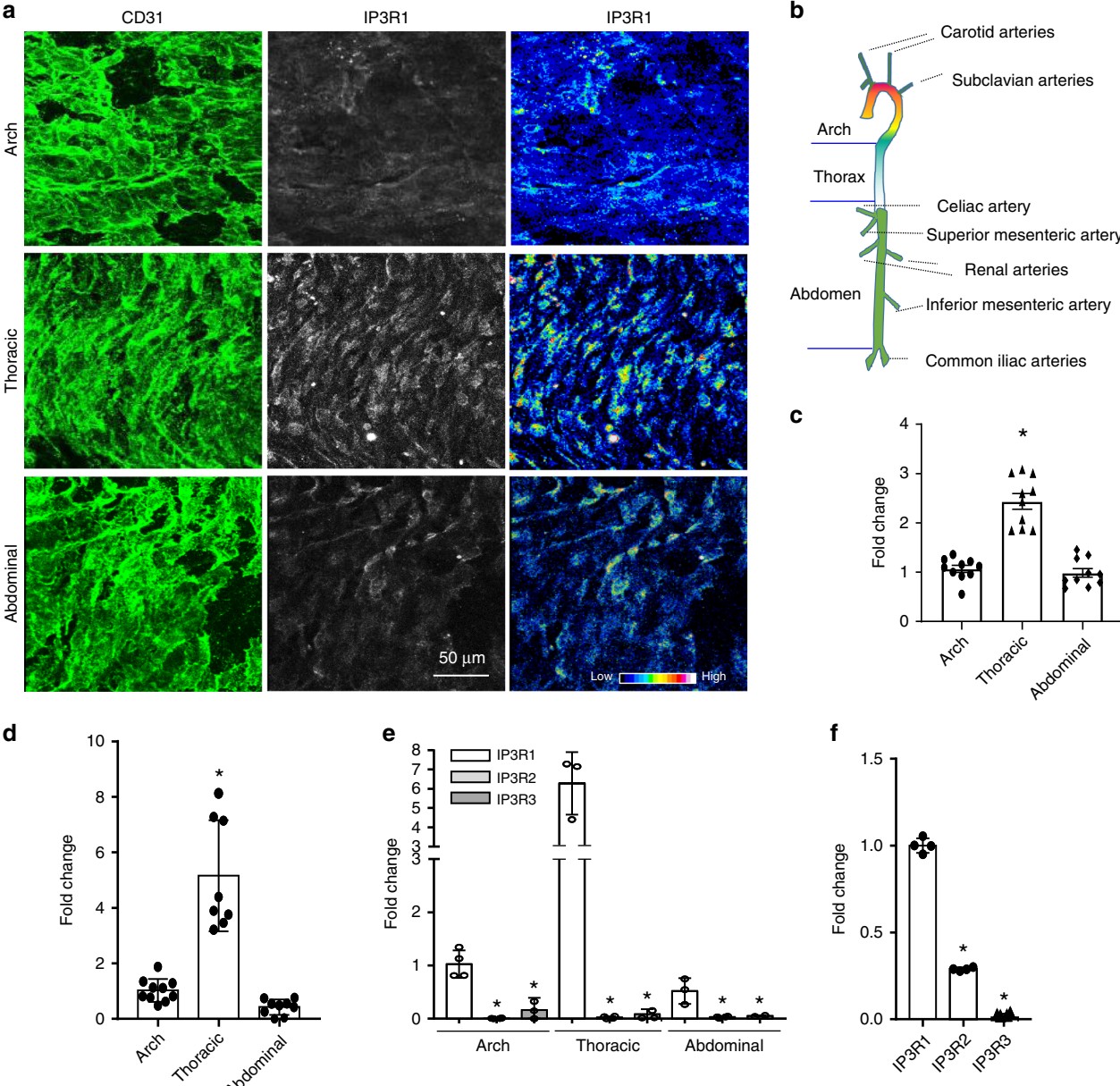

**Fig. 6 Abundance of IP3R1 in different regions of the aorta. a** Whole mount en face immunofluorescence staining, followed by confocal imaging. 0.5 μm optical sections of the endothelium were imaged. CD31 staining shows the endothelial cell surface and IP3R1 staining of the same optical sections is shown (left and center panels). The fluorescence intensity of IP3R1 staining of sections from the aortic arch, thoracic aorta, and abdominal aorta is also depicted (right panels). **b** A schematic diagram of the aorta highlighting the regions of analysis. **c** Quantification of IP3R1 immunofluorescent staining distribution in aortas (thoracic aorta vs. aortic arch and abdominal aorta, $n = 10$ microscopic fields captured for each segments, asterisk $P < 0.01$). **d** RT-PCR analysis of IP3R1 abundance in the aortic arch, thoracic aorta, and abdominal aorta (thoracic aorta vs. aortic arch or abdominal aorta, aortic RNA from seven mice was extracted for qPCR analysis, 4 repeats in each reaction, asterisk $P < 0.0001$). **e, f** Abundance of IP3R1, IP3R2, and IP3R3 mRNA in the aortic arch, thoracic aorta, and abdominal aorta. RNA was extracted from aortas of seven mice for qPCR analysis, and 4 repeats (**e**) in each reaction or in MAECs (**f**). (IP3R1 vs. IP3R2 and IP3R3, $n = 3$ independent experiments, asterisk $P < 0.0001$). All data were assessed using Student's $t$-test and are presented as the mean ± SEM.

endothelial cells decreases plasma nitric oxide production and increases blood pressure suggesting a functional redundancy in these calcium channels[46]. Interestingly, the same IP3 receptors regulate vascular smooth muscle contractility by increasing cytoplasmic calcium concentrations[47]. Accordingly, deletion of all three receptors in vascular smooth muscle cells significantly attenuates the increase in systolic blood pressure upon chronic infusion of angiotensin II, but failed to impact basal blood pressure[47]. IP3 receptors have also been implicated in playing an important role in atherosclerosis[28,29]. For instance, IP3R3 expression in human endothelial cells can be increased by

atheroprotective pulsatile shear stress (PS) downstream of KLF4[48]. In support of this finding, we found IP3R1 mRNA and protein levels were significantly higher under a pulsatile shear stress of 12 dynes/cm$^2$ vs. oscillatory shear stress of 1 dynes/cm$^2$ using human aortic endothelial cells (Fig. 7). We also discovered that IP3R1 distribution and abundance differed along the length of the aorta. This receptor was expressed to the greatest extent in the thoracic aorta, which is more resistant to atherosclerosis when compared to the aortic arch or abdominal aorta (Fig. 6). This indicates that differential shear stress affects IP3R1 expression, which is consistent with our in vitro experiments showing

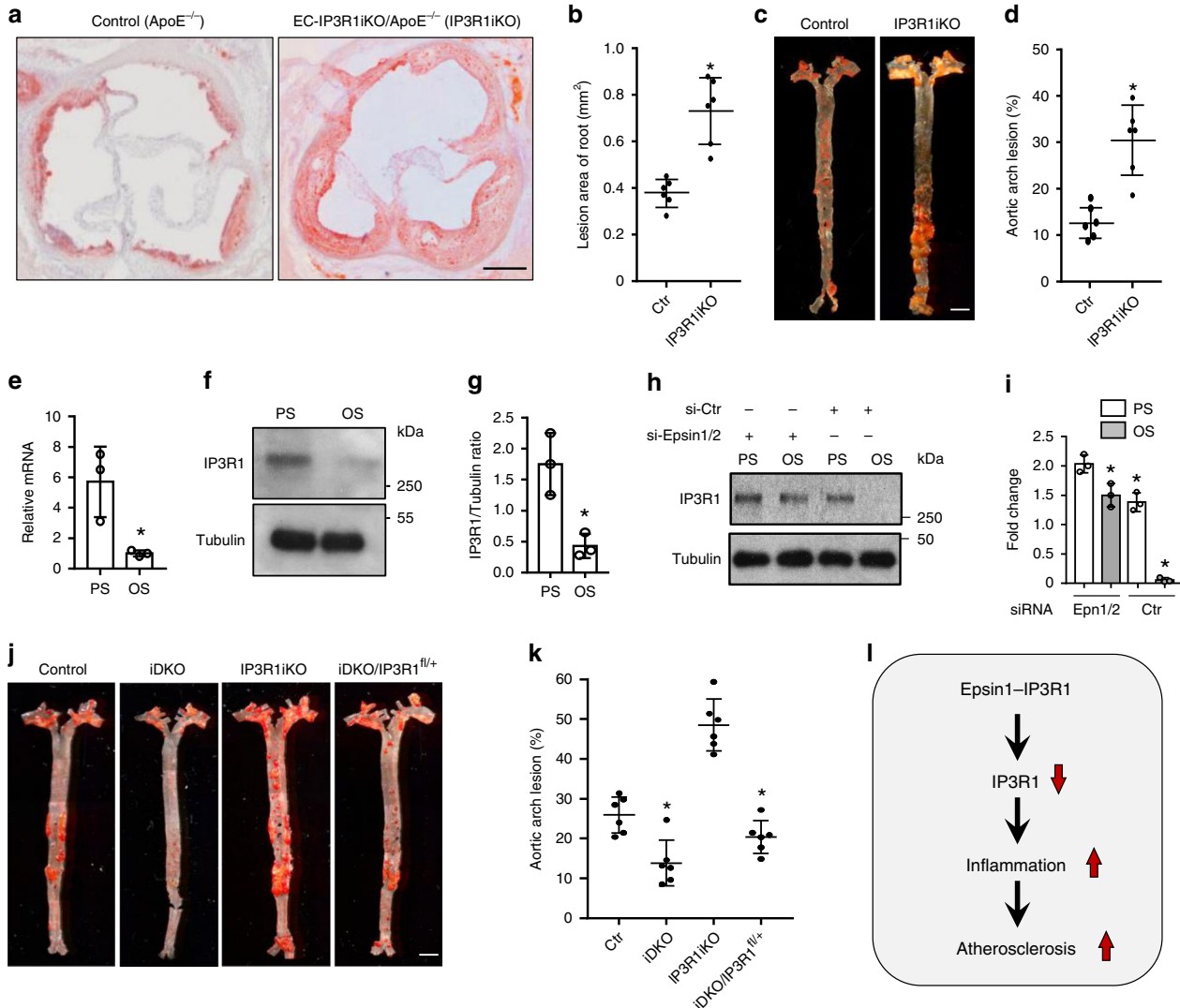

**Fig. 7 Reduction of endothelial IP3R1 accelerates atherosclerosis. a** to **d** ORO staining of atherosclerotic lesions in aortic roots (**a**) and arches (**c**) of control (ApoE$^{-/-}$) and EC-IP3R1iKO/ApoE$^{-/-}$ (IP3R1iKO) atherosclerotic mice fed a WD for 12–14 weeks. Statistical analysis of aortic roots and arches lesion are shown in (**b**, $n = 6$ aortic roots) and (**d**, $n = 6$ aortic arches), respectively (asterisk $P < 0.001$). Scale bars 500 μm (**a**) and 5 mm (**c**). **e–g** Human aortic endothelial cells (HAECs) subjected to $12 \pm 4$ dynes/cm$^2$ pulsatile shear stress (PS) or $1 \pm 4$ dynes/cm$^2$ oscillatory shear stress (OS) for 16 h. IP3R1 mRNA (**e**) and protein (**f**) levels were determined by qRT-PCR and western blot analysis, respectively. Quantitation of western blot results (**g**) (**e–g**, $n = 3$ independent repeats, asterisk $P < 0.001$). **h,i** HAECs subjected to PS or OS for 16 h in the presence of absence of epsin 1 and 2, followed by western blot (**h**) and statistical analyses (**i**) (epsin 1 vs. epsin 2, $n = 3$ independent repeats, asterisk $P < 0.05$; epsin 1 vs. Ctr, $n = 3$ independent repeats, asterisk $P < 0.003$; epsin 2 vs. Ctr., $n = 3$ independent repeats, asterisk $P < 0.005$;). **j, k** ORO staining of aortic arches of control, iDKO, IP3R1iKO, and iDKO/IP3R1$^{fl/+}$ mice fed a WD for 26–28 weeks (**j**). Scale bar 500 μm. Quantification of ORO staining showing rescue of atherosclerosis progression in iDKO/IP3R1$^{l/+}$ compared to iDKO mice (**k**) ($n = 6$ aortic arches in each group, asterisk $P < 0.001$). **l** Schematic diagram of endothelial epsin-mediated IP3R1 degradation and acceleration of inflammation and atherosclerosis. All data were assessed using Student's $t$-test and are presented as the mean ± SEM.

atheroprotective pulsatile laminar fluid flow upregulated IP3R1 and atheroprone oscillatory fluid flow downregulated this protein (Fig. 7). Together, these results indicate that increased IP3R1 levels may inhibit atherosclerotic lesion formation and progression.

Other groups have reported that IP3R1 protein levels and attendant calcium transients can be reduced in vascular smooth muscle cells by chronic oxLDL exposure, as well as in both rabbit and mouse atherosclerotic models[28,29]. In agreement with these reports, we found IP3R1 was reduced throughout the aortic wall (i.e., in the endothelium and smooth muscle layers) in atherosclerotic patients and in ApoE-null mice fed a western diet (Supplementary Fig. 5). Interestingly, oxidative stress also caused a decrease of IP3R1 through proteasomal degradation in cultured

smooth muscle cells resulting in reduced calcium efflux[49]. While these results suggest that IP3R1 may be involved in atheroma progression by disturbing cellular calcium homeostasis, our studies have connected endothelial cell epsin-mediated degradation of ubiquitinated IP3R1 to the progression of atherosclerosis in both cell culture and mouse model systems (Fig. 3).

Using a mutant mouse model, we show that the conditional deletion of IP3R1 expression in the endothelium promotes the pathogenesis of atherosclerosis (Fig. 7). In the aortic arch, loss of endothelial IP3R1 greatly increased lipid accumulation, particularly at the ascending and abdominal regions of the aortic arch. Our results demonstrate the atheroprotective effect of epsin deletion was the consequence of IP3R1 stabilization by prevention of proteasomal degradation. The combination of endothelial

epsin 1 and 2 deletions along with the genetic reduction of IP3R1 (EC-iDKO/EC-IP3R1$^{fl/+}$) on the ApoE-null background established the importance of the epsin-IP3R1 interaction in fueling atherosclerotic progression. As mentioned previously, we determined that epsin and IP3R1 interact via the ubiquitin interacting motif of epsins and the ubiquitinated SD domain of IP3R1 (Fig. 2). The latter domain is crucial for epsin-mediated IP3R1 degradation and inhibition of ubiquitination of IP3R1 using site-directed mutagenesis impaired binding of epsin 1 (Fig. 3). It is noted that IP3R1 ubiquitination residues are mainly located in the regulatory domain;[50] while we showed IP3R1 ubiquitination occurred at the SD site or the N-terminus, which is probably due to cell type and stimuli differences. In human atherosclerotic samples, we also found IP3R1 is downregulated, which implies that IP3R1 may be therapeutically-relevant. These experiments reveal that IP3R1 is degraded under pathological conditions and may provide a foundation for the development of a therapy to treat atherosclerosis.

We previously reported a specific role for epsins 1 and 2 in regulating VEGFR2 endocytosis and degradation[18,19,51]. The epsin-VEGFR2 interaction is also dependent on the epsin UIM and deletion of this domain abolishes binding of epsin to VEGFR2 in endothelial cells isolated from capillaries, veins, and small arteries[18,52]. It is thought that endothelial heterogeneity has contributed to differential function of many central regulators. In agreement with this assertion, we found that endothelial cells isolated from aortas express far less VEGFR2 compared to those isolated from small blood vessels (Supplementary Fig. 2), suggesting that the molecular events involved in angiogenesis and vessel sprouting are more frequent in small blood vessels (MBECs) vs. large blood vessels (MAECs). We speculate that targeting epsins in the atherosclerotic aortic endothelium would cause minimal disruption in epsin-dependent regulation of VEGFR2 stability given the much lower expression of this receptor in these aortic endothelial cells.

From a mechanistic perspective, we discovered that during chronic atherosclerosis, epsins modulate ER calcium release to fuel atherosclerosis by accelerating IP3R1 proteasomal degradation through the direct interaction of the epsin ubiquitin-interacting motif and the ubiquitinated IP3R1 suppressor domain. As we recently reported, deletion of macrophage epsins reduces atherosclerosis progression by preventing LRP-1, an anti-inflammatory protein involved in receptor-mediated endocytosis and protein degradation[22]. The question of whether endothelial epsins also interact with LRP-1 and target this protein for degradation is interesting and warrants further investigation.

Overall, our findings advance our understanding regarding the role of epsins in inflammation, intracellular calcium homeostasis, and atherosclerosis. Moreover, our results illuminate a dynamic interaction between plasma membrane proteins and an ER membrane protein; thereby, expanding our understanding of the role of epsins as adapter proteins for plasma membrane receptors and binding partners for intracellular organelle membrane receptors. Based on our animal and human data, IP3R1 appears to be essential in promoting atheroma progression in addition to its role in other diseases such as neurodegenerative disorders[53,54], breast cancer[55], and hypertension[56].

In summary, this study, along with our previous findings, supports the notion that both endothelial and macrophage epsins promote atherosclerosis and that loss of both endothelial and macrophage epsins prevents atherosclerosis[22]. Approaches that target both macrophage and endothelial epsins could provide an unprecedented benefit to simultaneously prevent endothelial dysfunction and foam cell formation to effectively halt atherosclerosis[57].

## Methods

**Human samples**. Human aortic arch samples were purchased from the Maine Medical Center BioBank (MMC BB). Medical information for atherosclerosis patient samples is provided in Supplementary Data file 3. Patient samples were initially collected by the MMC BB, which operates under an Institutional Review Board (IRB) approved protocol that is overseen by the Maine Medical Center Research Institute (MMCRI) Office of Research Compliance. The MMC BB IRB ensured that informed consent was sought from each prospective subject or the subject's legally authorized representative, in accordance with, and to the extent required by 45 CFR 46.116 and 21 CFR 50.20. The MMC BB IRB also covered the clinical analysis of the samples. Human Aortic Endothelial Cells (HAECs) were purchased from Cell Applications.

**Animals**. The mouse protocols in this study were approved by the Institutional Animal Care and Use Committee (IACUC) at Boston Children's Hospital and the Oklahoma Medical Research Foundation. Both male and female mice were used and housed using a 12/12-h light/dark cycle. Atherosclerotic ApoE-null mice (ApoE$^{-/-}$) were purchased from the Jackson Research Laboratory. The WD contained 1.3% cholesterol and 0.5% cholic acid (Atherogenic Rodent Diet TD.02028, Envigo Teklad).

Because epsin 1 and 2 double knockout mice (DKO) display embryonic lethality, we generated epsin 1$^{fl/fl}$; epsin 2$^{-/-}$ mice, which were subsequently combined with specific Cre deleter mouse strains to create tissue and cell type-specific inducible DKO (iDKO) mouse strains. The endothelial cell (EC)-specific deletion of epsin mouse strain was established by LoxP-Cre recombination where epsin 1 was flanked by LoxP sites in an epsin 2 null background[18,19]. A conditional deletion was created by breeding these mice with EC-specific Cre transgenic mice (iCDH5-Cre ERT2 or VE-cad-Cre ERT2) to create EC-iDKO mice. EC-iDKO mice were bred to ApoE$^{-/-}$ mice on a C57/BL6 background and backcrossed for seven generations and treated with 4-hydroxytamoxifen (5–10 mg/kg, body weight) 5 to 7 times every other day starting at age of 8 weeks of age[58]. Mice were subsequently fed a WD for the times indicated throughout the text and figures. For control mice, in addition to ApoE$^{-/-}$ mice, by taking advantage of the redundant function of epsins 1 and 2 (i.e., the lack of gross phenotype in single epsin KO mice) we also used littermate control Epsin 1$^{fl/fl}$; Epsin 2$^{-/-}$/ApoE$^{-/-}$ mice without tamoxifen-inducible Cre and treated these mice with 4-hydroxytamoxifen. In addition, we used Epsin 1$^{fl/fl}$; Epsin 2$^{-/-}$/ApoE$^{-/-}$ mice with tamoxifen-inducible Cre but without 4-hydroxytamoxifen treatment for additional controls. Collectively, we refer these control mice as ApoE$^{-/-}$ mice for simplicity. Importantly, we did not observe any significant difference among these control mice with respect to atherosclerotic plaque formation.

To generate IP3R1 LoxP-Cre mice, IP3R1 was flanked by LoxP sites and the EC-specific Cre mouse was bred with IP3R1$^{fl/fl}$ mice (iCDH5-Cre ERT2). The inducible EC-specific deletion of IP3R1; iCDH5-Cre ERT2 is denoted as IP3R1iKO, while the EC-specific IP3R1 heterozygous mice (EC-IP3R1$^{fl/+}$; iCDH5-Cre ERT2) is denoted as IP3R1$^{fl/+}$. IP3R1 mutants were then bred with ApoE-null mice on a C57/BL6 background, backcrossed and treated with 4-hydroxytamoxifen as described above. To examine atherosclerotic rescue in Epsin 1$^{fl/fl}$; Epsin 2$^{-/-}$/ApoE$^{-/-}$/iCDH5-Cre mice, IP3R1$^{fl/+}$ was introduced and Epsin 1$^{fl/fl}$; Epsin 2$^{-/-}$/IP3R1$^{fl/+}$/ApoE$^{-/-}$/iCDH5-Cre mice were obtained through breeding. For simplicity, we denote EC-iDKO/ IP3R1$^{fl/+}$/ApoE$^{-/-}$ mice as iDKO/ IP3R1$^{fl/+}$.

**Cell culture**. To isolate murine aortic endothelial cells (MAECs), aortas were collected and washed twice with PBS at 4 °C and then carefully stripped of fat and connective tissue. Aortas were cut into 3 mm long sections, and segments were put on Matrigel-coated (BD Biosciences) plate with EC medium. After 4 days, vascular networks were visible under the light microscope and tissue segments were removed. ECs were detached with Dispase II (Roche) at 5 mg/mL for 15–30 min. Cells were spun down and cultured in fresh EC medium containing FibrOut 11 (CHI Scientific) to remove fibroblasts. The identity of isolated ECs was confirmed by immunofluorescent staining using EC markers CD31, vWF, and VEGFR2 antibodies and an anti-α-SMA antibody was used as a negative control. A full list of reagents including antibodies, peptides, and primers is included in the supplementary information (Supplementary Data file 4). Cultured MAECs were treated with 5 μM tamoxifen for 3 days to induce the deletion of epsins or IP3R1 gene from EC-iDKO/ApoE$^{-/-}$ or EC-IP3R1$^{fl/fl}$: iCDH5-Cre mice, MAECs isolated from ApoE$^{-/-}$ mice as controls that also were treated with 5 μM tamoxifen at the same time. Cells were treated with 100 μg/mL oxLDL, 500 μg/mL cholesterol crystal, or 50 μM 7-KC at different times as indicated. MAECs were also isolated from WT mice and treated with 100 μM leupeptin, 10 μM salubrinal, and 2.5 μM MG132 in the presence or absence of 100 μg/mL oxLDL for 36 h. In addition, we isolated these cells from ULK$^{-/-}$ mice to test whether autophagy can block oxLDL-mediated IP3R1 degradation. The preparation of these reagents is described below. Murine brain endothelial cells (MBECs) were isolated and cultured as described previously[13,18]. HEK 293T cells were cultured according to the supplier's directions (ATCC). Primary endothelial cells isolated from mice were used within two passages.

**Mass spectrometry**. MAECs were isolated as described previously and above[59]. For the epsin 1 mass spectrometry, ten aortas were used to produce approximately 100 million cells using MAECs within two passages. MAECs were treated with 100 µg/mL oxLDL for 30 min, followed by immunoprecipitation with an epsin-1 antibody in rec-protein-G-Sepharose 4B beads (Life Technologies). Beads were thoroughly washed with IP buffer for 5 times. Samples were analyzed using SDS-PAGE (Perfect NT Gel W, 10–20% acrylamide, 28 wells; DRC Co.) according to the manufacturer's protocol. The gels were stained with Coomassie Brilliant Blue R250 (Sigma-Aldrich). Protein bands were excised, destained in 50% acetonitrile containing 50 mM $NH_4HCO_3$, and then washed with deionized water. Gels were dehydrated in 100% acetonitrile for 15 min and dried in a micro-centrifugal vacuum concentrator (Thermo Fisher Scientific) for 60 min. These were immersed for 45 min at 4 °C in 10 to 30 mL of 50 mM Tris (pH 9.0) containing 0.1 ng/mL trypsin (Promega). Excess trypsin was discarded and the gel pieces were stored for 24 h at 37 °C in a minimal volume (10 to 20 mL) of 50 mM Tris (pH 9.0) buffer. The digested peptide fragments from the gel pieces were diffused into the surrounding solution over a 24 h period. The solution was then slowly transferred to 1.5 mL siliconized plastic tubes and stored at 4 °C. To further recover the peptide fragments that remained in the gel pieces, 20 min incubations using minimal volumes of 5% formic acid containing 50% ACN were performed at room temperature. The solutions containing peptides were pooled in siliconized tubes. The tryptic sample was diluted 5-fold with 0.1% formic acid, and desalted using MonoSpin C18 (GL Sciences) followed by lyophilization. The lyophilized sample was dissolved in Invitrosol (Thermo Fisher Scientific) to enhance the recovery of hydrophobic peptides[60].

Peptide digests were analyzed with the LTQ Orbitrap Discovery system (Thermo Fisher Scientific) coupled with Nanospace SI-2 HPLC (Osaka-Soda), and separated on a CAPCELL PAK $C_{18}$ MG III-H S3 reverse phase column (Shiseido). The flow rate of the mobile phase was 200 µL/min. The solvent composition of the mobile phase was programmed to change after 24 min with variable mixing ratios of solvent A (water-formic acid = 100:0.005, v/v) to solvent B (water-ACN-formic acid = 10:90:0.005, v/v/v): 0% B (0–2.8 min), 0–10% B (2.8–3.0 min), 10–25% B (3.0–9.0 min), 25–40% B (9.0–19.0 min), 40–60% B (19.0–21.0 min), 60–95% B (21.0–21.1 min), and 95% B (21.1–24 min). A spray voltage of 5000 V was applied.

The Mascot version 2.2.6 search engine (Matrix Science) was used to identify proteins from the mass and tandem mass spectra of peptides. Peptide data was matched by searching the UniProtKB human database using the MASCOT engine. Database search parameters were: peptide mass tolerance, 3 ppm; fragment tolerance, 0.8 Da; enzyme was set to trypsin, allowing up to one missed cleavage; fix modifications, carbamidomethyl (cysteine); variable modifications, oxidation (methionine). Scaffold version 3.6.1 (Proteome Software Inc.) was used to validate MS/MS based peptide and protein identifications. Peptide identifications were accepted if they could be established at greater than 95.0% probability as specified by the Peptide Prophet algorithm[61]. Protein identifications were accepted if they could be established at greater than 99.0% probability and contained at least one identifiable peptide. Protein probabilities were assigned by the Protein Prophet algorithm[62]. Quantitative analysis of protein abundance was determined by using a spectral count method[63].

**Neutrophil/macrophage adhesion assays**. Neutrophils were isolated as described earlier[64,65]. Briefly, mice were injected intraperitoneally with 1 mL of 4% thioglycollate medium. After 4 h (3-day thioglycollate induction for macrophage isolation), mice were sacrificed using isoflurane and, under sterile conditions, peritoneal fluid was collected using a 10 mL syringe with a 20 G needle. Cells were spun down and analyzed for purity by flow cytometry (based on scatter and the expression of Ly6G).

Adhesion of isolated wild-type neutrophils (or macrophages) to the surface of MAECs under flow was performed as previously described[64,65]. Neutrophils ($1 \times 10^6$/mL in HBSS with $Ca^{2+}$, $Mg^{2+}$, and 0.5% HSA) were perfused over TNFα-stimulated (50 ng/mL for 3 h) monolayers of MAECs on 35 mm dishes mounted in a parallel-plate flow chamber (Glycotech) at a shear stress of 1 dynes/cm². After 10 min, images were collected using a video microscopy system for 5 min and analyzed for adherent cells using Element software (Nikon). Adherent cells were counted from at least five different fields in each experiment and each experiment was repeated more than three times.

The static adhesion assay was performed as described with some modifications[66] Isolated MAECs were seeded at a density of $8 \times 10^4$ per well using a 12-well plate containing a coverslip. After reaching confluency, cells were stimulated with 50 ng/mL TNFα for 3 h to induce expression of adhesion molecules. Freshly isolated mouse neutrophils from a GFP-labeled transgenic mouse donor were added to the MAECs at $8 \times 10^5$ per well and allowed to adhere for 15 min, followed by three washes with Hank's balanced salt solution (HBSS) to remove non-adherent neutrophils, and stained with DAPI (Thermo Fisher Scientific). The number of bound neutrophils were imaged by fluorescence microscopy and the ratio of adherent to total cells was expressed as fold change.

**Macrophage migration assays**. The macrophage migration assay was performed as previously described[67]. Briefly, MAECs from WT and EC-iDKO mice were added to the upper chamber membrane of Transwell inserts (Corning) overnight. The next day, confluent MAECs were treated with TNFα (50 ng/mL) or LPS

(200 ng/mL) for 3 h. Peritoneal macrophages were then harvested by lavage of the peritoneal cavity using 6 mL of sterile PBS 3 days after intraperitoneal injection of 4% thioglycollate (TG; 1 mL/animal). These macrophages were added on the top of the endothelial cells in the Transwell insert. After 3 h, macrophage migration through the Transwell membrane was detected by brightfield microscopy by imaging the lower chamber of the Transwell plate.

**Serum glucose and lipid analyses**. Serum glucose was measured using a glucometer (OneTouch). Triglyceride levels were determined by a kit (BioVision and Thermo Fisher Scientific), followed by spectrophotometric quantification. Total cholesterol, HDL, and LDL/VLDL were measured using a kit (Abcam).

**Staining of atherosclerotic lesions**. Oil Red O (ORO) staining of the aortic root and aortic arch was performed as described previously[37,68]. In brief, after being fed a WD for the indicated number of weeks, mice were fasted for 14 h, exsanguinated, and perfused with PBS. The heart and aortic tissues were fixed in 4% paraformaldehyde for 16 h. To analyze lesions, hearts were dissected from aortas, embedded in tissue freezing medium and sectioned (10 µm thickness). Four serial sections over a distance of 200 µm were collected from each mouse and stained with ORO and counterstained with hematoxylin. For analyzing lesions in the aortic arch, the intimal surface was exposed by a longitudinal cut from the ascending arch to 5 mm distal of the left subclavian artery to allow the lumen of the aortic arch to be laid flat. The aorta was rinsed for 5 min in 80% and 100% propylene glycol, respectively, stained with ORO for 8–10 min at 65 °C, and destained in 80% propylene glycol for 5 min, and then rinsed with distilled water. Digital images of the aorta were captured with a stereomicroscope (Olympus) and the lesion area was quantified from the aortic arch to 5 mm distal of the left subclavian artery using NIH Image J.

**En face confocal microscopy**. Aortas were perfused with PBS and 4% PFA and isolated (i.e., from the heart to the renal aorta) as described above and then incubated for 20 min in 4% PFA[22]. Whole mount immunofluorescence staining was performed based on a previous study[69]. After quenching using 100 mM glycine-PBS (pH 7.4), tissues were permeabilized, washed, and blocked with PBS containing 5% donkey serum, 1% BSA, 75 mM NaCl, 18 mM $Na_3C_6H_5O_7$, and 0.01% Triton X-100 for 2 h at room temperature. Aortas were then incubated with primary antibodies to IP3R1 (1:100), α-SMA-Cy3 (1:500), and PECAM1 (1:100) overnight. Following a 2 h incubation with donkey anti-rat or anti-rabbit Alexa Fluor conjugated secondary antibodies, tissues were mounted and imaged using a confocal microscope (LSM 880, Zeiss) at 488, 550, and 680 nm. Regions-of-interest (ROI) (i.e., aortic arches, thoracic aortas, and abdominal aortas) were imaged using 0.5 µm Z axis increments. Endothelial cells were identified as PECAM positive and α-SMA negative. Specific regions were subjected to high magnification imaging after low magnification tile scanning of the entire aorta. Endothelial cells in each ROI demonstrated morphologies consistent with local shear stress conditions. Image reconstruction and analyses were performed using Zen Black software (Zeiss) and Image J[70,71].

**In vitro shear stress experiments**. Human aortic endothelial cells (HAECs) were cultured in M199 medium supplemented with 15% fetal bovine serum (Hyclone), 1 ng/mL recombinant human endothelial growth factor (Sigma-Aldrich), 90 µg/mL heparin sodium (Sigma-Aldrich), 100 U/mL streptomycin/penicillin (Hyclone), and 100 U/mL sodium pyruvate (Hyclone). A circulating flow system was used to impose shear stress on confluent monolayers of HAECs seeded on glass slides as described[72]. A reciprocating syringe pump connected to the circulating system introduced a sinusoidal (1 Hz) component onto the shear stress. The atheroprotective pulsatile shear flow (PS) or atheroprone oscillatory shear flow (OS) generated shear stresses of 12 ± 4 or 1 ± 4 dynes/cm², respectively. The flow system was enclosed in a chamber held at 37 °C and ventilated with 95% humidified air plus 5% $CO_2$.

**Flow cytometry**. Flow cytometry was performed essentially as described previously[73]. In short, MAECs ($1 \times 10^5$) were incubated at 4 °C for 30 min in 100 µL of PBS plus 1% bovine serum albumin (BSA) with a PE-conjugated anti-mouse VE-cadherin antibody, washed three times, and analyzed by flow cytometry (Becton Dickinson). PE-conjugated mouse IgG1 (R&D Systems) was used as an isotype control. Data were analyzed using FlowJo version 10 software (Tree Star).

For analysis of resident immune cells in aortas, cells were isolated from aortas as described earlier[6,74]. In brief, mice were anesthetized and perfused with PBS and perivascular adipose tissue was removed. Aortas were minced into small pieces and digested with an enzymatic solution containing 125 U/mL collagenase type XI, 60 U/mL hyaluronidase type I-s, 60 U/mL DNase I and 450 U/mL collagenase type I in PBS containing 20 mM HEPES at 37 °C for 3 h. After filtering through a 70 µm filter, cells were re-suspended in FACS buffer, and incubated with Fc-blocking antibody (eBioscience) for 15 min on ice before being stained with specific antibodies. The antibodies used were as follows: FITC-CD45, PE/Cy7-CD11b, APC/Cy7-CD11c, PE-CD19, Alex Flour-700-TCR-b and Pacific blue-Ly6-C (all were obtained from BioLegend and used at 1:100 dilution). Cells were simultaneously stained with propidium iodide. After washing, immunofluorescence

was detected using an LSR II (BD Biosciences) and data were analyzed using FlowJo (Tree Star) software.

**Cloning and transfection**. Epsin 1 plasmids were constructed as previously described[18,20]. IP3R1 plasmids IP3R1HAWT, IP3R1 HAΔ1-1581, IP3R1 HAΔ1-1903, and IP3R1 HAΔ1-2268 were a kind gift from Dr. Richard J.H. Wojcikie-wicz[75]. Truncated expression constructs of the N-terminal domain (NTD) and regulatory domain (RD), suppressor domain (SD), IP3 binding cores α and β (IBC), SD plus the IBCα and IBCβ domains were created by PCR amplification and insertion into the pcDNA3.1 vector (primer information can be found in Supplementary Data file 4). Double mutation (K126R/K129R) and triple mutation (K126R/K129R/K143R) IP3R1 constructs were made using the QuikChange II Site-directed mutagenesis kit (Agilent) according to the manufacturer's directions and confirmed by DNA sequencing. Plasmids were transfected into HEK 293T cells for 24 h using Lipofectamine 2000 Transfection Reagent (Thermo Fisher Scientific) and cell lysates were used for immunoprecipitations or western blot analyses. Transfection of constructs into MAECs was performed using an Amaxa Nucleo-fector™ II Device (Lonza). In brief, $1 \times 10^6$ MAECs were mixed with 5 μg plasmids and 100 μL Amaxa Basic Nucleofector Kit-Primary Endothelial Cells Solution (Lonza), followed by an immediate pulse using program A-034. Cells were cultured up to 3 days for gene expression analyses.

**Other experimental procedures**. Immunoprecipitation, western blotting, H&E staining, immunofluorescence staining, confocal microscopy, and cell culture maintenance was performed according to standard published methodologies[18,20,37,52,68,76]. Western blots were repeated at least 3 times and the precise numbers of repetitions are indicated in the figure legends. Confocal and fluorescence microscope images were captured using an Ix81 Spinning Disc Confocal Microscope (Olympus) with a plan apochromat 60x objective (Olympus) and Orca-R[2] Monochrome Digital Camera (Hamamatsu).

Oxidized LDL (oxLDL) was prepared as described earlier[68]. Final preparations were dialyzed extensively against PBS buffer. Cholesterol crystals were made according to a previous literature[77]. Cholesterol (Sigma-Aldrich) was dissolved in 95% ethanol (12.5 g/l) and heated to 60 °C, filtered through filter paper (Whatman) while still warm, and left at room temperature to allow crystallization to proceed. The crystals were collected by filtering, autoclaved, ground using a sterile mortar and a pestle and stored at −20 °C until use. Endotoxin (EU/mg) was monitored using the LAL kit (Pierce).

A calcium mobilization assay in MAECs was performed by U-Pharm Laboratories LLC (Parsippany, NJ)[32]. The cell plate was decanted and the medium was replaced with 20 μL HBSS Hank's Balanced Salt Solution (HBSS) (138 mM NaCl, 5 mM KCl, 1.3 mM CaCl2, 0.5 mM MgCl2, 0.4 mM MgSO4, 0.3 mM KH2PO4, 0.3 mM Na2HPO4, 5.6 mM glucose). Fluo-8 calcium dye (U-Pharm Laboratories) was added at 20 μL per well to achieve 2 μM final concentration and then incubate at 37 °C for 60 min followed by incubation at room temperature for an additional 30 min. The baseline fluorescence was measured using the FDSS6000 system (Hamamatsu) with optical filters for an excitation wavelength of 490 nm and emission of 525 nm. During the same measurement, cells were stimulated with ATP (0, 0.0003, 0.003, 0.1, 0.3, 1.0, 3.3, 30.3, and 100 μmol/L) at room temperature and the fluorescence signal was recorded at 1.5 s intervals over a period of 3 min. Ca$^{2+}$ mobilization results were expressed as either peak fluorescent changes or ratios over background.

**Gene ontology analysis**. For gene ontology (GO) analyses, the list of proteins identified from mass spectrometry was imported to generic GO term finder at http://go.princeton.edu/ and GO terms were searched against the mouse (*M. musculus*) database. Ontology aspects including process, function, and components were analyzed. The ratio of the percentage of individual GO terms in the protein list to that in the whole mouse genome was calculated and defined as the fold-change of enrichment. Fold-changes of enriched GO terms were plotted with their P-values calculated by the GO algorithm.

**Statistical analyses**. Data were presented as the mean ± SEM. Data were analyzed by a two-tailed Student's *t*-test or ANOVA with appropriate post hoc tests as needed. A *P*-value of less than 0.05 were considered statistically significant. All animal experiments consist of five or more mice in each group and represent a minimum of three repetitions. All in vitro experiments were repeated at least five times. Data used for statistical analyses and uncropped Western blots can be found in the Source Data folder.

**Reporting summary**. Further information on research design is available in the Nature Research Reporting Summary linked to this article.

## Data availability

All data generated or analyzed supporting the findings of this study are available within the paper and its supplementary information files. All data are available from the corresponding author upon reasonable request. Lesion plaques for aortic roots and arches were quantified using Image J available from NIH website (https://imagej.nih.gov/ij/).

Statistical analysis was conducted using GraphPad Prism software (https://www.graphpad.com/scientific-software/prism/). Source data are provided with this paper.

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

## Acknowledgements

We are grateful to Dr. Ju Chen at the University of California, San Diego, for providing IP3R1 LoxP-Cre mice. This work was supported, in part, by NIH grants R01HL093242, R01HL118676, R01HL130845, R01HL133216, R01HL137229, and the American Heart Association (AHA) Established Investigator Award to H.C.; the AHA Scientist Development Grant (SDG) 12SDG8760002, as well as OCAST grants AR11-043 and HR14-056 to Y.D.; and the AHA SDG grant 17SDG33630161 to K.S.

## Author contributions

Y.D. and H.C. initiated and designed the study. Y.D. performed experiments and analyzed data. Y.K., H.M., and Y.K. performed the mass spectrometry. Y.L. performed whole mount staining of aortas and blood pressure measurements. M.H. performed the parallel plate shear stress experiments and analyses. T.Y. performed the in vitro rolling assay. X.L. analyzed atherosclerotic patient samples for epsin expression. K.C. conducted the macrophage migration analyses. K.O. helped make the IP3R1 LoxP-Cre mice. R.J.H.W. provided IP3R1 vectors and made constructive comments. K.C., A.W., L.Y., and M.L.B.

performed tissue staining (IF, ORO, IHC). K.L. performed the immune cell analysis from aortic arches. J.W.-S. and J.B. performed FACS analysis of MAECs for VE-Cadherin. G.C., H.L., and Y.L. conducted the bioinformatics analyses. S.S. measured the biochemical parameters of blood. S.W. provided technical support, including mouse breeding, mouse genotyping, and mouse colony maintenance. All other colleagues were involved in technical support and discussion of the data. Y.D., D.B.C., and H.C. wrote the manuscript.

## Competing interests

The authors declare no competing interests.
