## [Peer Review File · Nature Communications]

Reviewers' Comments:

Reviewer #1:

Remarks to the Author:

In this study by Dong and colleagues, the role of endothelial Epsin-1 in the development of atherosclerosis was investigated. The authors identified an interaction of Epsin-1 with the endoplasmic reticulum protein IP3R1, which regulates calcium release in the cytosol. Using isolated endothelial cells, the authors found that binding of Epsin-1 to IP3R1 led to a proteolytic degradation of IP3R1, which was mediated by the proteasome. Mapping studies using recombinant constructs with deleted domains pinpointed the interaction sites to the ubiquitin-interacting motif in Epsin-1 and the suppressor domain in IP3R1. Subsequent experiments revealed that ubiquitination of IP3R1 precedes Epsin-1 binding and degradation. The absence of Epsin-1 (and -2) maintained calcium homeostasis in endothelial cells treated with oxidized lipoproteins. In addition, whereas wild-type endothelial cells showed a strong upregulation of adhesion molecules after stimulation with TNF α , Epsin-deficiency showed a suppression of this upregulation after TNF-stimulation. After this extensive in vitro characterization, mice with genetic deletions of endothelial Epsin-1 and -2 and of endothelial IP3R1 (or combinations thereof) were investigated in a diet-induced model of atherosclerosis. Deletion or reduction of IP3R1 was found to accelerate atherosclerosis, while deletion of Epsins-1 and -2 led to a notable reduction in plaque formation. Deletion of a single IP3R1 allele on an Epsin-1 and -2 double endothelial knockout background partially restored atherosclerotic lesion formation to the wild-type situation.

This is an interesting study, in which the role of the endothelium in the development of atherosclerosis is highlighted. Endothelium dysfunction, associated with disturbed intracellular calcium homeostasis, drives atherosclerotic plaque formation. This study links the Epsin-1-mediated degradation of IP3R1 to endothelial dysfunction during plaque formation. Although of interest, a number of issues need to be addressed to increase impact and to resolve conceptual unclarities.

1. Endothelial cells can be considered to be an integral part of the immune system, as they should be permissive to leukocyte (trans-)migration. Since the proteasomal targeting of IP3R1 by Epsin-1 appear to be a physiologic function (of Epsin 1), how would general inflammation and the immune response be affected by (functional) Epsin-1/2 deficiency? This is a relevant question to this reviewer for 2 reasons: (i) The authors advocate the identified function of Epsin-1 to be a valuable drug target. (ii) The Epsin-1-mediated degradation of IP3R1 appears to occur during inflammation, irrespective of the trigger (oxidized lipoproteins, TNF α). A remark to the latter is that in animal models, it is unclear which is the trigger for IP3R1 ubiquitinylation, (diet-induced) hyperlipidemia, systemic inflammation, aberrant flow conditions, or all of those.

The study would be strengthened if the authors could show and discuss whether:

- Epsin-1 mediated endothelial dysfunction is a patho(physio)logic mechanism that relates more or less exclusively to vascular inflammation and atherogenesis.
- Endothelial calcium homeostasis (e.g. artificially modulated by BAPTA-AM) closely relates to adhesion molecule expression and leukocyte trafficking, or whether a different downstream effect is responsible for this.
- the increased influx of different subsets of inflammatory cells into the plaque is directly related to a supposed increased permissiveness of the endothelium when Epsins are present (can be tested by transmigration through different genotypes of cultured cells).

2. A further question that arose during reading the manuscript is related to flow conditions. Although already touched down upon later in the manuscript (using cultured human endothelial cells under physiologic and pathologic flow conditions), it would be interesting to expand these experiments to those performed with the cultured mouse cells. The study would be conceptually coherent when aberrant flow conditions would lead to similar Epsin-1-directed degradation of IP3R1 as triggering with cytokines or oxidized lipoproteins. The authors might also relate IP3R1 expression to the different regions of the aorta or carotid artery (e.g. inner vs outer curvature and abdominal, or carotid bifurcation).

3. Endothelial IP3R1 is ubiquitinated under "abnormal" conditions. Do the authors have any mechanistic explanation why IP3R1 expression and calcium homeostasis should be altered under these conditions? And how this occurs?

Other remarks:

As the study was done on a double Epsin-1/2 background, please discuss the implications for human disease more elaborately (where both epsins are present). Similar counts for the different IP3 receptors.

Please more clearly indicate the statistical significance between groups in figure 6i.

Reviewer #2:

Remarks to the Author:

Dong, Chen and colleagues studied the role for Epsin 1 and 2 in atherosclerosis. They report that Epsin1 interacts with the inositol 1,4,5-trisphosphate receptor type I (IP3R1), only under atherogenic conditions (following stimulation with high doses of oxLDL, Cholesterol or 7-KC). They showed that the interaction is Ubiquitin-dependent and occurs between the UIM domain of Epsin 1 and the suppressor domain of IP3R1. Some of the sites of Ub on IP3R1 were mapped to K126, 129 and 143. They claimed that such interaction triggers proteasomal degradation of the channel. The rest of the manuscript (Fig. 4, 5 and 6) reports calcium levels and atherosclerosis phenotypes in Epsin or IP3R1 KO mice but is rather phenomenological and could just be circumstantial (affecting Ca²⁺ levels will have many pleiotropic effects).

The interaction of Epsin1 and IP3R1 and the degradation of the latter under atherogenic conditions is new and interesting. However, the interpretation of the data for the molecular mechanism is very flawed and unlikely to be true. The data do not support the conclusion that such degradation is mediated by the proteasome. The authors need to do a rigorous investigation of the mechanism and provide direct evidence and they need to rule out alternative hypotheses.

1) IP3R1 is a 6-pass transmembrane protein. How could the proteasome extract the transmembrane parts to degrade it? Transmembrane proteins are degraded in the lysosome (following endocytosis or autophagy), not by the proteasome.

The only evidence the authors got to support proteasome degradation is the use of MG132.

But this is not an evidence of proteasomal degradation. It is well known that blocking the proteasome by MG132 (or by any proteasomal drugs such as ALLN, Lactacystin, Bombesin etc..) induces several compensatory effects: it triggers unfolded protein response (UPR), which activates autophagy (reviewed 10 years ago in PubMed ID (PMID)20040365 and more recently in PMID30333975), and it decreases the availability of free Ub (thus indirectly slowing down endosomal sorting and lysosomal degradation).

In addition some of the drugs may have side effects: MG132 was reported to affect transcription (PMID30647455 amongst others) and ALLN is also a Cathepsin inhibitor and thus will block Lysosomal degradation as well (PMID8087844).

Compensatory autophagy following MG132 (or other proteasome inhibitors) is mediated by the IRE1 arm of the UPR (IRE1 etc..) and JNK1, which phosphorylates Bcl-2, thereby disrupting autophagy-inhibitory interaction with Beclin-1 (reviewed in PMID20040365).

Interestingly, IP3R1 interacts with Bcl-2 (PMID15613488 and PMID19706527) and is known to be involved in autophagy (PMID30251688, PMID22082873, PMID23565295, PMID28254579). Moreover, Epsin is involved in autophagy in flies (PMID19305132).

Furthermore, Ubiquitination of IP3R1 was found to be as much K48 (proteasome+autophagy) as

K63 (Lysosomes), with K63 found to accumulate most rapidly least 40% of Ub was Mono-Ub (PMID18955483), which suggests an autophagic and/or lysosomal degradation, the proteasome needs poly-Ub)

Finally, Lysosomal inhibition (NH₄Cl and Chloroquine) blocked IP3R1 degradation (PMID9139693, despite what the authors of this paper said: Fig. 7 clearly showed that IP3R1 was at 100% of control levels upon AngII).

Thus, the authors must rule out alternative hypotheses such as autophagy, ER-phagy and Lysosomal degradation.

- The role for autophagy can be easily ruled out by depleting Atg5, Atg7 and p62, the use of autophagy inducers and inhibitors (serum starvation, rapamycin, resveratrol, 3MA and chloroquine).
 - Lysosomal degradation can be inhibited by Bafilomycin (more specific than NH₄CL) and Leupeptides or other Cathepsin inhibitors.
 - UPR can be tested by GPR78 and phospho-eIF2 α levels and inhibited by Salubrinal, IRE1a and ATF4 KD.
- Both the OxLDL and the MG132 effects should be tested upon autophagy, lysosomal and UPR inhibitions.

The authors also ought to back up their claim that IP3R1 is degraded by the proteasome by providing direct evidence and mechanistic insights. How do Epsin actually link IP3R1 to the proteasome? Etc..

2) Fig. 1c, 1e, 1g, 3c, 3h and 3j: it is not clear that Ub is actually on IP3R1 (its Mw did not change). How did the authors rule out that the Ub is not on another protein being co-IP? I understand the K to R mutations, but this is on ectopically expressed proteins.

3) Fig. 3j is apparent contradiction with Fig.3a: no Epsin1 and 2 increase upon oxLDL. Unless this is an issue with alignment Epsin1 and 2 blots on 3a?

4) Reference 28 does not support line 98-100 (it is about Ca²⁺, not stress)

Reviewer #3:

Remarks to the Author:

This is a study which makes the novel observation that the endocytic adaptor proteins epsin are linked to the ER Ca²⁺ channel IP3R1 in endothelial cells and that this attachment is related to IP3R1 degradation by the Ub/proteasome pathway. The authors show that perturbation of this Ca²⁺ signaling pathway affects the susceptibility of endothelium to atherogenic stimuli. The experiments are a nice combination of cell culture and animal models. The results are well organized and the experiments are performed in a thorough manner. In my opinion, the basic manuscript is worthy of publication in Nature communications and should be of interest to a wide audience. Despite my generally positive view, I do have a number of specific concerns that the authors should be encouraged to address. These are detailed below:

- 1) The authors provide data using mutant constructs that the site of interaction with IP3R1 is the suppressor domain and that interaction involves 3 lysines in this domain at K126, K129, K143. Wojcikiewicz and coworkers have characterized the Ub binding sites in all 3 IP3R isoforms and none of the sites were in the N-terminal portion of the IP3R. Although different experimental systems were used, the authors should reference these studies and provide an explanation for the discrepancies. Wojcikiewicz et al have also shown that the IP3Rs are ubiquitinated with both K48 and K63 chains. Have the authors looked to see what kind of chains are being formed on the SD?
- 2) Previous studies by the Wojcikiewicz lab have identified the adaptor proteins erlin1 & 2 and the ring-finger E3 ligase RNF170 as being key to the ER associated degradation of IP3Rs. How are the epsin proteins integrated into this system? Are they mediating a completely separate degradation mechanism or are they alternative adaptors that plug in to the erlin/RNF170 complexes?
- 3) As the authors state in their discussion the endothelial cells have all 3 IP3R isoforms which contribute to the Ca²⁺ signal and presumably all 3 isoforms are downregulated in response to oxLDL or 7-KC. Yet, the emphasis of the paper is on IP3R1 and loss of just the IP3R1 gene or restoration of one allele of IP3R1 is sufficient to increase/lower the number of plaques seen in the animal models fed a WD diet. Does this mean that IP3R2 and IP3R3 have no role in the endothelial cells and IP3R1 has some selective effect. How is the Ca²⁺ signal effected in MAECs derived from these animal models?
- 4) Although the source of the animal models are mentioned in the supplementary table of materials, the origin of the IP3R1 floxed mice is not clear.
- 5) Which specific endocytic proteins are being referred to on p6L91? Is this info in the supplementary Table?
- 6) The authors state that there are SNPs in IP3R1 linked to increase risk of cardiovascular disease. They go on to suggest that these result in "gain or loss of function". What is the evidence for this functional measurements and are the SNPs in the coding sequence of IP3R1?
- 7) The panel (f) in Figure 2 is not well described in the text. If this panel uses just the receptor's NTD and RD then presumably the cartoon hosing these constructs in (g) should precede (f). Things may be clearer if the construct # in the cartoon is also used in the labeling of (h).
- 8) In supplemental Figure 7 the abbreviation "N-glyc" is not defined. What is the purpose of showing the PTMs and why is this shown for only aa1-53? It is unclear why the lysine 129 position is highlighted but not the other potential Ub sites.
- 9) The series of experiments involving identifying interacting sites use HEK293T cells. The legend to Fig 3 includes a sentence to indicate that the experiments involved a 30min pretreatment with oxLDL to induce the Ub of the IP3R1 or constructs. The sentence should be included in the main text. It is not clear from the experiments that the treatment with oxLDL is absolutely necessary to see Ub of IP3R1 or to observe the interaction with the epsins since none of the experiments have a control from which the oxLDL treatment has been omitted. It should be noted that Wojcikiewicz and coworkers have proposed that the Ub/proteasome pathway may play a role in the basal turnover of IP3Rs.

Responses to the Reviewers' Comments

Reviewer #1 (Remarks to the Author):

In this study by Dong and colleagues, the role of endothelial Epsin-1 in the development of atherosclerosis was investigated. The authors identified an interaction of Epsin-1 with the endoplasmic reticulum protein IP3R1, which regulates calcium release in the cytosol. Using isolated endothelial cells, the authors found that binding of Epsin-1 to IP3R1 led to a proteolytic degradation of IP3R1, which was mediated by the proteasome. Mapping studies using recombinant constructs with deleted domains pinpointed the interaction sites to the ubiquitin-interacting motif in Epsin-1 and the suppressor domain in IP3R1. Subsequent experiments revealed that ubiquitination of IP3R1 precedes Epsin-1 binding and degradation. The absence of Epsin-1 (and -2) maintained calcium homeostasis in endothelial cells treated with oxidized lipoproteins. In addition, whereas wild-type endothelial cells showed a strong upregulation of adhesion molecules after stimulation with TNF α , Epsin-deficiency showed a suppression of this upregulation after TNF-stimulation. After this extensive in vitro characterization, mice with genetic deletions of endothelial Epsin-1 and -2 and of endothelial IP3R1 (or combinations thereof) were investigated in a diet-induced model of atherosclerosis. Deletion or reduction of IP3R1 was found to accelerate atherosclerosis, while deletion of Epsins-1 and -2 led to a notable reduction in plaque formation. Deletion of a single IP3R1 allele on an Epsin-1 and -2 double endothelial knockout background partially restored atherosclerotic lesion formation to the wild-type situation.

This is an interesting study, in which the role of the endothelium in the development of atherosclerosis is highlighted. Endothelium dysfunction, associated with disturbed intracellular calcium homeostasis, drives atherosclerotic plaque formation. This study links the Epsin-1-mediated degradation of IP3R1 to endothelial dysfunction during plaque formation. Although of interest, a number of issues need to be addressed to increase impact and to resolve conceptual unclarity.

Reply: We appreciated your positive comments and suggestions.

1. Endothelial cells can be considered to be an integral part of the immune system, as they should be permissive to leukocyte (trans-)migration. Since the proteasomal targeting of IP3R1 by Epsin-1 appear to be a physiologic function (of Epsin 1), how would general inflammation and the immune response be affected by (functional) Epsin-1/2 deficiency? This is a relevant question to this reviewer for 2 reasons: (i) The authors advocate the identified function of Epsin-1 to be a valuable drug target. (ii) The Epsin-1-mediated degradation of IP3R1 appears to occur during inflammation, irrespective of the trigger (oxidized lipoproteins, TNF α). A remark to the latter is that in animal models, it is unclear, which is the trigger for IP3R1 ubiquitinylation, (diet-induced) hyperlipidemia, systemic inflammation, aberrant flow conditions, or all of those.

Reply: We consider the high level of epsin expression in atherosclerosis fundamental to epsin-mediated IP3R1 ubiquitination and degradation. Despite that, we agree with the reviewer that it is

unclear in the animal models exactly what triggers epsin upregulation. We have found epsins are upregulated by oxLDL treatment (Fig. 3) and this results in an interaction with IP3R1 (Figs. 1, 2). In addition, new data (Figs. 6 and 7) shows that laminar shear stress upregulates IP3R1 *in vivo* and *in vitro*. Furthermore, in the absence of epsins 1 and 2, IP3R1 is relatively abundant under laminar and disturbed flow conditions (Fig 7h, i). Interestingly, disturbed flow (oscillatory shear stress) decreased IP3R1 faster in the presence of epsins 1 and 2, which further suggests that epsins accelerate IP3R1 degradation. However, the question of whether inflammation and immune responses lead to epsin upregulation and whether they share similar molecular mechanisms is currently unclear and warrants further investigation.

The study would be strengthened if the authors could show and discuss whether epsin-1 mediated endothelial dysfunction is a patho(physio)logic mechanism that relates more or less exclusively to vascular inflammation and atherogenesis.

Reply: To address this issue, we treated MAECs with TNF α and LPS to stimulate inflammatory signaling in the presence or absence of epsins in mouse aortic endothelial cells (MAECs). We found that loss of epsins inhibited inflammatory pathways (Fig. 4). As the reviewer noted, endothelial calcium homeostasis (artificially modulated by BAPTA-AM) is closely related to adhesion molecule expression and leukocyte trafficking. Epsins downregulate IP3R1 expression pathologically, causing intracellular calcium reduction (Fig. 4) and disturbed intracellular homeostasis. As it is well known that a main feature of calcium imbalance is ER stress, we tested if ER stress signaling is altered in the absence of epsins in MAECs. Our results show that loss of epsins in endothelial cells, significantly downregulated ER stress signaling; however, we feel inclusion these studies are beyond the scope of this manuscript and we hope to address this issue in an upcoming publication.

The reviewer also noted that the increased influx of different subsets of inflammatory cells into the plaque is directly related to a supposed increased permissiveness of the endothelium when epsins are present. At the reviewer's suggestion, we have used Transwell migration assays to test this hypothesis using WT and epsin-deficient MAECs. Our data showed that loss of epsins reduces macrophage transmigration (Fig 4). Using HUVECs treated with siRNA and the human monocyte cell line U-937, we have obtained the similar results (not shown). These findings are also in line with our *in vitro* rolling data (Fig. 4).

2. A further question that arose during reading the manuscript is related to flow conditions. Although already touched down upon later in the manuscript (using cultured human endothelial cells under physiologic and pathologic flow conditions), it would be interesting to expand these experiments to those performed with the cultured mouse cells. The study would be conceptually coherent when aberrant flow conditions would lead to similar Epsin-1-directed degradation of IP3R1 as triggering with cytokines or oxidized lipoproteins. The authors might also relate IP3R1 expression to the different regions of the aorta or carotid artery (e.g. inner vs outer curvature and abdominal or carotid bifurcation).

Reply: Due to a technical limitation, MAECs do not adhere well to the rolling chamber under shear stress. Consequently, we used Human Aortic Endothelial Cells (HAECs) and siRNA to downregulate epsin1 and 2 expression. We performed shear stress experiments using HAECs in the presence or absence of epsins, and analyzed IP3R1 expression. Our data shows that pulsatile shear (PS) upregulates IP3R1 (Fig. 7), while oscillatory shear (OS) downregulates IP3R1 (Fig. 7). Whole mount staining of aortas and *en face* confocal scanning of the endothelium showed that more IP3R1 protein was identified in the thoracic aorta when compared to the aortic arch or descending aorta (Fig. 6a to c). To elaborate on these findings, aortic regions were isolated and used for RNA extraction (see Fig. 6b). RT-PCR analyses demonstrated that a much high level of IP3R1 was apparent in the thoracic aorta compared to the aortic arch and abdominal regions (Fig. 6d), which was in line with the results presented in Fig. 6a and c. These data suggested that IP3R1 is increased in atherosresistant regions, which are largely subjected to pulsatile shear stress, when compared to atheroprone regions, which are primarily characterized with disturbed hemodynamic forces such as oscillatory shear stress. Our findings imply that increased IP3R1 levels are atheroprotective.

3. Endothelial IP3R1 is ubiquitinated under "abnormal" conditions. Do the authors have any mechanistic explanation why IP3R1 expression and calcium homeostasis should be altered under these conditions? And how this occurs?

Reply: We speculate that elevated cholesterol levels in the ApoE^{-/-} mouse model fed a western diet would be the culprit. Cholesterol, especially oxidized LDL is known to produce oxidative stress that directly impacts the ER. It is likely that IP3R1 is misfolded or subjected to other posttranslational modifications in these conditions, which certainly warrants future investigation. This would result in degradation of IP3R1 through ERAD (ER Associated Degradation) as a protein quality control mechanism, which disrupts calcium homeostasis and implies that epsins play an important role in this process.

Other remarks:

As the study was done on a double Epsin-1/2 background, please discuss the implications for human disease more elaborately (where both epsins are present). Similar counts for the different IP3 receptors.

Reply: We have now discussed these points in our revised manuscript.

Please more clearly indicate the statistical significance between groups in figure 6i.

Reply: We have more clearly marked the statistical significance.

Reviewer #2 (Remarks to the Author):

Dong, Chen and colleagues studied the role for Epsin 1 and 2 in atherosclerosis. They report that Epsin1 interacts with the inositol 1,4,5-trisphosphate receptor type I (IP3R1), only under atherogenic conditions (following stimulation with high doses of oxLDL, Cholesterol or 7-KC). They showed that the interaction is Ubiquitin-dependent and occurs between the UIM domain of Epsin 1 and the suppressor domain of IP3R1. Some of the sites of Ub on IP3R1 were mapped to K126, 129 and 143. They claimed that such interaction triggers proteasomal degradation of the channel. The rest of the manuscript (Fig. 4, 5 and 6) reports calcium levels and atherosclerosis phenotypes in Epsin or IP3R1 KO mice but is rather phenomenological and could just be circumstantial (affecting Ca²⁺ levels will have many pleiotropic effects).

The interaction of Epsin1 and IP3R1 and the degradation of the latter under atherogenic conditions is new and interesting. However, the interpretation of the data for the molecular mechanism is very flawed and unlikely to be true. The data do not support the conclusion that such degradation is mediated by the proteasome. The authors need to do a rigorous investigation of the mechanism and provide direct evidence and they need to rule out alternative hypotheses.

Reply: Thank you for your comments and suggestions. We agree that further investigation of the degradative mechanisms would elevate the impact and significance of our study. These control experiments are important to the molecular mechanism identified in this report.

1) IP3R1 is a 6-pass transmembrane protein. How could the proteasome extract the transmembrane parts to degrade it? Transmembrane proteins are degraded in the lysosome (following endocytosis or autophagy), not by the proteasome.

Reply: While it is true that transmembrane proteins are normally degraded in the lysosome, it is also well documented that IP3R1 degradation occurs through ER-associated degradation (ERAD)^{1,2}, an ubiquitination-proteasome system. To eliminate concerns from this reviewer, we have tested other protein degradation systems as suggested.

The only evidence the authors got to support proteasome degradation is the use of MG132.

But this is not an evidence of proteasomal degradation. It is well known that blocking the proteasome by MG132 (or by any proteasomal drugs such as ALLN, Lactocystin, Bombesin etc..) induces several compensatory effects: it triggers unfolded protein response (UPR), which activates autophagy (reviewed 10 years ago in PubMed ID (PMID) 20040365 and more recently in PMID30333975), and it decreases the availability of free Ub (thus indirectly slowing down endosomal sorting and lysosomal degradation).

In addition some of the drugs may have side effects: MG132 was reported to affect transcription (PMID30647455 amongst others) and ALLN is also a Cathepsin inhibitor and thus will block Lysosomal degradation as well (PMID8087844).

Compensatory autophagy following MG132 (or other proteasome inhibitors) is mediated by the IRE1 arm of the UPR (IRE1 etc..) and JNK1, which phosphorylates Bcl-2, thereby disrupting autophagy-inhibitory interaction with Beclin-1 (reviewed in PMID20040365).

Interestingly, IP3R1 interacts with Bcl-2 (PMID15613488 and PMID19706527) and is known to be involved in autophagy (PMID30251688, PMID22082873, PMID23565295, PMID28254579). Moreover, Epsin is involved in autophagy in flies (PMID19305132).

Furthermore, Ubiquitination of IP3R1 was found to be as much K48 (proteasome+autophagy) as K63 (Lysosomes), with K63 found to accumulates most rapidly least 40% of Ub was Mono-Ub (PMID18955483), which suggests an autophagic and/or lysosomal degradation, the proteasome needs poly-Ub)

Finally, Lysosomal inhibition (NH4Cl and Chloroquine) blocked IP3R1 degradation (PMID9139693, despite what the authors of this paper said: Fig. 7 clearly showed that IP3R1 was at 100% of control levels upon AngII).

Reply: We appreciated the potential confounding effects presented by the reviewer and we have added this information to our Results and Discussion where appropriate.

Thus, the authors must rule out alternative hypotheses such as autophagy, ER-phagy and Lysosomal degradation.

- The role for autophagy can be easily ruled out by depleting Atg5, Atg7 and p62, the use of autophagy inducers and inhibitors (serum starvation, rapamycin, resveratrol, 3MA and chloroquine).

Reply: We have performed new experiments to better distinguish between autophagy, lysosomal degradation, and UPR in IP3R1 degradation as suggested by the reviewer. In particular, we have isolated WT, Atg1/Ulk-deficient MAECs and test the oxLDL and MG132 effects on IP3R1 degradation. Our data now shows that in MAECs the loss of ULK1 does not affect IP3R1 degradation (Supplemental Fig. 3)

- Lysosomal degradation can be inhibited by Bafilomycin (more specific than NH4CL) and Leupeptin or other Cathepsin inhibitors.

Reply: We have performed experiments to test the oxLDL and the MG132 effects on IP3R1 degradation in the presence of the lysosome inhibitor, Leupeptin. Our data shows that lysosome inhibition does not affect IP3R1 stabilization (Supplemental Fig. 3).

- UPR can be tested by GPR78 and phospho-eIF2alpha levels and inhibited by Salubrinal, IRE1a and ATF4 KD.

Reply: We have performed experiments to test the oxLDL and the MG132 effects on IP3R1 degradation in the presence of the UPR inhibitor, Salubrinal. Our data show that UPR inhibitor, Salubrinal has little effect IP3R1 stabilization (Supplemental Fig. 3).

Both the OxLDL and the MG132 effects should be tested upon autophagy, lysosomal and UPR inhibitions.

The authors also ought to back up their claim that IP3R1 is degraded by the proteasome by providing direct evidence and mechanistic insights. How do Epsin actually link IP3R1 to the proteasome?

Reply: To better understand this question, we conducted a new experiment to test how epsins are linked to IP3R1 degradation. IP3R1 degradation has been suggested to involve the RNP170 E3 ligase complex. Epsins bind to IP3R1 to accelerate degradation of this receptor. In our Co-IP experiments, we found that epsins bind to erlin 1 and 2 as well as RNF170 to promote ER-mediated IP3R1 proteasomal degradation in the ERAD (Fig. 3m).

2) Fig. 1c, 1e, 1g, 3c, 3h and 3j: it is not clear that Ub is actually on IP3R1 (its Mw did not change). How did the authors rule out that the Ub is not on another protein being co-IP? I understand the K to R mutations, but this is on ectopically expressed proteins.

Reply: IP3R1 is a huge protein with 2749 amino acids (migrating at ~320 kDa in a SDS-PAGE gel); so, migratory changes due to relatively small levels of ubiquitination of the protein is very difficult to discern by gel migration. On the other hand, our data shows the migration of IP3R1 in gels is consistent through the IP and when subjected to denaturation-renaturation and testing with an anti-ubiquitin antibody (not shown). This implies that any binding partners were removed, and IP3R1 is the ubiquitinated protein. (not shown). This implies that any binding partners were removed, and IP3R1 is the ubiquitinated protein.

3) Fig. 3j is apparent contradiction with Fig.3a: no Epsin1 and 2 increase upon oxLDL. Unless this is an issue with alignment Epsin1 and 2 blots on 3a?

Reply: Our quantification shows oxLDL treatment increased epsin 1 and 2 more than 2 times as shown in figure 3J. We have now repeated these experiments and presented new blots.

4) Reference 28 does not support line 98-100 (it is about Ca²⁺, not stress)

Reply: We have deleted this reference.

Reviewer #3 (Remarks to the Author):

This is a study which makes the novel observation that the endocytic adaptor proteins epsin are linked to the ER Ca²⁺ channel IP3R1 in endothelial cells and that this attachment is related to IP3R1 degradation by the Ub/proteasome pathway. The authors show that perturbation of this Ca²⁺ signaling pathway affects the susceptibility of endothelium to atherogenic stimuli. The experiments are a nice combination of cell culture and animal models. The results are well organized and the experiments are performed in a thorough manner. In my opinion, the basic manuscript is worthy of publication in Nature Communications and should be of interest to a wide audience.

Reply: We thank the reviewer for the positive comments and for the support of our work.

Despite my generally positive view, I do have a number of specific concerns that the authors should be encouraged to address. These are detailed below:

1) The authors provide data using mutant constructs that the site of interaction with IP3R1 is the suppressor domain and that interaction involves 3 lysines in this domain at K126,K129,K143. Wojcikiewicz and coworkers have characterized the Ub binding sites in all 3 IP3R isoforms and none of the sites were in the N-terminal portion of the IP3R. Although different experimental systems were used, the authors should reference these studies and provide an explanation for the discrepancies. Wojcikiewicz et al have also shown that the IP3Rs are ubiquitinated with both K48 and K63 chains. Have the authors looked to see what kinds of chains are being formed on the SD?

Reply: We believe that cell type and stimuli play important roles in driving ubiquitination of different sites. Our results strongly suggest that the putative polyubiquitin sites are located at the part of the SD domain immediately following the first 53 aa (*i.e.* 54-223 aa region of SD or SDA53). This prediction is in agreement with a previous study that implied the polyubiquitin sites reside at the N-terminus of IP3R1³. As suggested by the Wojcikiewicz group, IP3R1 degradation occurs through K48 polyubiquitination. Consequently, we have tested the specific Ub chains on the SD by using K48 or K63 antibodies. We overexpressed IP3R1 (with a HA-tag) in HEK293 cells in the presence of MG132 and oxLDL, and immunoprecipitated proteins with HA antibody, followed by immunoblotting with K48 and K63 antibodies, respectively. We demonstrated that the K48 antibody gives a band equivalent to IP3R1 (by HA antibody blotting), while the K63 antibody did not produce a band (Fig.3n and o). These results mean that IP3R1 degradation occurs through K48 polyubiquitination under oxLDL conditions.

2) Previous studies by the Wojcikiewicz lab have identified the adaptor proteins erlin1 &2 and the ring-finger E3 ligase RNF170 as being key to the ER associated degradation of IP3Rs. How are the epsin proteins integrated into this system? Are they mediating a completely separate degradation mechanism or are they alternative adaptors that plug in to the erlin/RNF170 complexes?

Response: We have tested whether epsin binds to erlin1&2 or RNF170 to promote ER-mediated IP3R1 proteasomal degradation. Our results show that epsin 1 is associated with the RNF170 complex, composed of RNF170, erlin 1, and erlin 2 (Fig. 3). Please also see our reply to reviewer 2's question concerning IP3R1 degradation.

3) *As the authors state in their discussion the endothelial cells have all 3 IP3R isoforms which contribute to the Ca²⁺ signal and presumably all 3 isoforms are downregulated in response to oxLDL or 7-KC. Yet, the emphasis of the paper is on IP3R1 and loss of just the IP3R1 gene or restoration of one allele of IP3R1 is sufficient to increase/lower the number of plaques seen in the animal models fed a WD diet. Does this mean that IP3R2 and IP3R3 have no role in the endothelial cells and IP3R1 has some selective effect? How is the Ca²⁺ signal effected in MAECs derived from these animal models?*

Reply: We have tested the expression levels of three IP3R isoforms in endothelial cells and found that IP3R1 is the predominantly expressed isoform in MAECs. We also show that IP3R2 is about 20-25% of IP3R1, and IP3R3 is not expressed at a measurable level (Fig. 6). These findings are in line with a previous publication⁴. We wanted to know if epsin modulates IP3R2 and IP3R3 *in vivo*. Consequently, we analyzed these three isoforms by RT-PCR in aortic samples, and our results show that IP3R1 remains the predominant isoform (Fig. 6e). It is also true that IP3R1 is the predominant isoform in mouse primary cultured aortic endothelium cells (MAECs) (Fig. 6f), in line with a previous result (see the newly-added reference 39 in the manuscript).

4) *Although the source of the animal models are mentioned in the supplementary table of materials, the origin of the IP3R1 floxed mice is not clear.*

Reply: We now show detailed IP3R1-floxed mice information in Supplemental Fig. 10 and have included a new reference (39).

5) *Which specific endocytic proteins are being referred to on p6L91? Is this info in the supplementary Table?*

Reply: We were referring to clathrin and epsin 2. We have now clarified the text to reflect this information.

6) *The authors state that there are SNPs in IP3R1 linked to increase risk of cardiovascular disease. They go on to suggest that these result in "gain or loss of function". What is the evidence for this functional measurement and are the SNPs in the coding sequence of IP3R1?*

Reply: Supplemental Figure 4 has been removed per the Editor's request.

7) *The panel (f) in Figure 2 is not well described in the text. If this panel uses just the receptor's NTD and RD then presumably the cartoon hosing these constructs in (g) should precede (f). Things may be clearer if the construct # in the cartoon is also used in the labeling of (h).*

Reply: We have adjusted the text and/or legend accordingly.

8) *In supplemental Figure 7 the abbreviation “N-glyc” is not defined. What is the purpose of showing the PTMs and why is this shown for only aa1-53? It is unclear why the lysine 129 position is highlighted but not the other potential Ub sites.*

Reply: We apologize for the confusion and have removed the putative site labeling and improved the figure.

9) *The series of experiments involving identifying interacting sites use HEK293T cells. The legend to Fig 3 includes a sentence to indicate that the experiments involved a 30min pretreatment with oxLDL to induce the Ub of the IP3R1 or constructs. The sentence should be included in the main text. It is not clear from the experiments that the treatment with oxLDL is absolutely necessary to see Ub of IP3R1 or to observe the interaction with the epsins since none of the experiments have a control from which the oxLDL treatment has been omitted. It should be noted that Wojcikiewicz and coworkers have proposed that the Ub/proteasome pathway may play a role in the basal turnover of IP3Rs.*

Reply: We have compared the Ub of IP3R1 and the interaction of IP3R1 with epsins with or without oxLDL treatment and found no significant differences between these conditions. Regardless, we have now corrected the sentence in the main text.

REFERENCES

- 1 Lu, J. P., Wang, Y., Sliter, D. A., Pearce, M. M. & Wojcikiewicz, R. J. RNF170 protein, an endoplasmic reticulum membrane ubiquitin ligase, mediates inositol 1,4,5-trisphosphate receptor ubiquitination and degradation. *J Biol Chem* **286**, 24426-24433, doi:10.1074/jbc.M111.251983 (2011).
- 2 Wright, F. A. & Wojcikiewicz, R. J. Chapter 4 - Inositol 1,4,5-Trisphosphate Receptor Ubiquitination. *Prog Mol Biol Transl Sci* **141**, 141-159, doi:10.1016/bs.pmbts.2016.02.004 (2016).
- 3 Bhanumathy, C. D., Nakao, S. K. & Joseph, S. K. Mechanism of proteasomal degradation of inositol trisphosphate receptors in CHO-K1 cells. *J Biol Chem* **281**, 3722-3730, doi:10.1074/jbc.M509966200 (2006).
- 4 Yuan, Q. *et al.* Maintenance of normal blood pressure is dependent on IP3R1-mediated regulation of eNOS. *Proc Natl Acad Sci U S A* **113**, 8532-8537, doi:10.1073/pnas.1608859113 (2016).

Reviewers' Comments:

Reviewer #1:

Remarks to the Author:

The authors have sufficiently revised the manuscript and I have no further comments.

Reviewer #2:

Remarks to the Author:

My main concern was the absence of direct evidence for the molecular mechanism mediating the degradation of a multi-pass channel (IP3R1) by the proteasome. The only evidence linking the degradation to the receptor was the use of proteasome inhibitors, but these have many pleiotropic effects, including Ub availability and indirect effects on autophagy and lysosomal degradation.

I appreciate the effort by the authors to address some of my concerns. Unfortunately, the new experiments on Figure S3 are not rigorous as they lack basic controls to check that their treatments (Leupeptin and salubrinal) actually worked, and so are inconclusive at this stage.

Likewise, the ULK KO only blocks general autophagy (although here too, no controls were done), not selective ER-phagy, see PMID31100386 for a recent review. So this experiment did not rule out ER-phagy and the concern still remains.

The new experiment finding that Epsins bind to the Erlin1-2 and RNF170 complex is a good addition to the paper but is not a proof that IP3R1 is degraded by the proteasome. It is just the identification of an E3 ligase that may ubiquitinate it (the importance of RNF170 was only inferred, was not tested in the present manuscript).

The argument by the authors that there is a precedent of the IP3R1 to be degraded by the proteasome during the ERAD response is very weak for 2 reasons:

1. the ERAD response handles misfolded or damages protein within the lumen of the ER, followed by retro-translocation into the cytosol and then ubiquitin-dependent degradation by the proteasome. How this applies to a mature, fully folded and membrane embedded receptor (IP3R1) upon recognition by cytosolic proteins (Epsins), is totally unclear to me.

2. the papers cited by the authors (Lu JP et al JBC 2011) and the rest of the literature on ERAD-mediated degradation of IP3R1 (PMID12421829, 16103111, 10839985, 19751772 etc.) suffers from exactly the same flaws than the present study as the only evidence ever linking the degradation of IP3R1 to the proteasome was the use of MG132, ALLN etc.. thus the concern this is an indirect effect blocking another mechanism is still valid.

In conclusion, I am sorry to say that the authors have not convinced me much more. I appreciate their effort to address my comments but their experiments either did not address some of my comments or were not rigorous enough to rule out other explanations or provide direct evidence for the molecular mechanism.

However, my disagreement concerns only a small part of the study and it should not distract us from the quality and novelty of the rest of the manuscript. I do not wish to block the publication of the study but, in absence of direct evidence or clear molecular mechanism, I encourage the authors to consider revising their claim about IP3R1 being degraded by the proteasome in this set up. Perhaps being more nuanced about the possibility of alternative mechanisms (ER-phagy etc.) would be wise at this stage.

Reviewer #3:

Remarks to the Author:

The authors have addressed my comments with regard to a) K48/63 sites of Ub linkage; b) role of RNF170; c) role of other IP3R isoforms and d) role of other IP3R isoforms. They have also cleared up a number of minor points that I raised regarding the text.

I do not think the authors have adequately dealt with my point that the SD is the site of Ub being different from what is known from proteomic analysis of Ub attachment sites in IP3R1. The authors' response to reviewers makes the valid point that this could just be due to the use of a different cell type. This should be inserted as a sentence in the paper.

Some other comments

1) Wojcikiewicz has proposed that the Ub/proteasomal degradation of IP3Rs can be viewed as a response of the cell to chronic elevation of IP3. Do the atherogenic stimuli like oxLDL increase IP3?

2) It would be useful to show the coomassie gel lane for the IgG control used in the mass-spec studies. Similarly, are the list of proteins included for the controls in the supplementary table 2?

3) It is made clear at the end of the discussion that the epsin is in the PM and the IP3R is in the ER. However, the consequences of this different localization of the proteins is not discussed in relation to the data or the model. Are the events being studied by the authors occurring at contact points between the PM and ER.

4) The discussion on p16 is repetitive and can be shortened.

Responses to the Reviewers

We were pleased that our manuscript was again received favorably by the Reviewers of *Nature Communications* and we are now submitting a re-revised manuscript. We found the Reviewers comments to be informative, and we have revised the paper based on their recommendations. New data has been incorporated into the paper as detailed below. Changes to the text are indicated in red and new data has been added to both the figures and supplemental figures.

Reviewers' comments:

Reviewer #2 (Remarks to the Author):

My main concern was the absence of direct evidence for the molecular mechanism mediating the degradation of a multi-pass channel (IP3R1) by the proteasome. The only evidence linking the degradation to the receptor was the use of proteasome inhibitors, but these have many pleiotropic effects, including Ub availability and indirect effects on autophagy and lysosomal degradation.

I appreciate the effort by the authors to address some of my concerns. Unfortunately, the new experiments on Figure S3 are not rigorous as they lack basic controls to check that their treatments (Leupeptin and salubrinal) actually worked, and so are inconclusive at this stage.

Likewise, the ULK KO only blocks general autophagy (although here too, no controls were done), not selective ER-phagy, see PMID31100386 for a recent review. So this experiment did not rule out ER-phagy and the concern still remains.

The new experiment finding that Epsins bind to the Erlin1-2 and RNF170 complex is a good addition to the paper but is not a proof that IP3R1 is degraded by the proteasome. It is just the identification of an E3 ligase that may ubiquitinate it (the importance of RNF170 was only inferred, was not tested in the present manuscript).

The argument by the authors that there is a precedent of the IP3R1 to be degraded by the proteasome during the ERAD response is very weak for 2 reasons:

1. the ERAD response handles misfolded or damaged protein within the lumen of the ER, followed by retro-translocation into the cytosol and then ubiquitin-dependent degradation by the proteasome. How this applies to a mature, fully folded and membrane embedded receptor (IP3R1) upon recognition by cytosolic proteins (Epsins), is totally unclear to me.

2. the papers cited by the authors (Lu JP et al JBC 2011) and the rest of the literature on ERAD-mediated degradation of IP3R1 (PMID12421829, 16103111, 10839985, 19751772 etc.) suffers from exactly the same flaws than the present study as the only evidence ever linking the degradation of IP3R1 to the proteasome was the use of MG132, ALLN etc.. thus the concern this is an indirect effect blocking another mechanism is still valid.

In conclusion, I am sorry to say that the authors have not convinced me much more. I appreciate their effort to address my comments but their experiments either did not address some of my comments or were not rigorous enough to rule out other explanations or provide direct evidence for the molecular mechanism.

However, my disagreement concerns only a small part of the study and it should not distract us from the quality and novelty of the rest of the manuscript. I do not wish to block the publication of the study but, in absence of direct evidence or clear molecular mechanism, I encourage the authors to consider revising their claim about IP3R1 being degraded by the proteasome in this set up. Perhaps being more nuanced about the possibility of alternative mechanisms (ER-phagy etc.) would be wise at this stage.

Reply: We appreciate your helpful comments regarding IP3R1 degradation, and we believe that a thorough investigation is required to fully elucidate the molecular mechanism. We hope to publish a follow-up paper on this matter; however, these studies are beyond the scope of this study. Consequently, we have adjusted our text to more accurately reflect our findings and the limitations of our study. Of note, we have included two endothelial cell-specific KO mice in this paper (epsins DKO and IP3R1 KO), which have never been published before, and our results may provide clues for the eventual development of therapies for atherosclerosis.

Reviewer #3 (Remarks to the Author):

The authors have addressed my comments with regard to a) K48/63 sites of Ub linkage; b) role of RNF170; c) role of other IP3R isoforms and d) role of other IP3R isoforms. They have also cleared up a number of minor points that I raised regarding the text.

I do not think the authors have adequately dealt with my point that the SD is the site of Ub being different from what is known from proteomic analysis of Ub attachment sites in IP3R1. The authors' response to reviewers makes the valid point that this could just be due to the use of a different cell type. This should be inserted as a sentence in the paper.

Reply: This is added to the Discussion.

Some other comments

1) Wojcikiewicz has proposed that the Ub/proteasomal degradation of IP3Rs can be viewed as a response of the cell to chronic elevation of IP3. Do the atherogenic stimuli like oxLDL increase IP3?

Reply: oxLDL may transiently increase IP3 levels.

2) It would be useful to show the Coomassie gel lane for the IgG control used in the mass-spec studies. Similarly, are the list of proteins included for the controls in the supplementary table 2?

Reply: IgG control of Coomassie-stained gel lanes and IgG control mass spec data are included in Supplementary Data.

3) It is made clear at the end of the discussion that the epsin is in the PM and the IP3R is in the ER. However, the consequences of this different localization of the proteins is not discussed in relation to the data or the model. Are the events being studied by the authors occurring at contact points between the PM and ER.

Reply: this has been included in the Discussion.

4) The discussion on p16 is repetitive and can be shortened.

Reply: we have revised the Discussion.